# APML: Adaptive Probabilistic Matching Loss for Robust 3D Point Cloud Reconstruction

**Sasan Sharifipour[1], Constantino Álvarez Casado[1], Mohammad Sabokrou[2], Miguel Bordallo López[1]**
[1]Center for Machine Vision and Signal Analysis (CMVS), University of Oulu (Finland)
[2]Okinawa Institute of Science and Technology (OIST), Japan
`sasan.sharifipour,constantino.alvarezcasado,miguel.bordallo@oulu.fi`

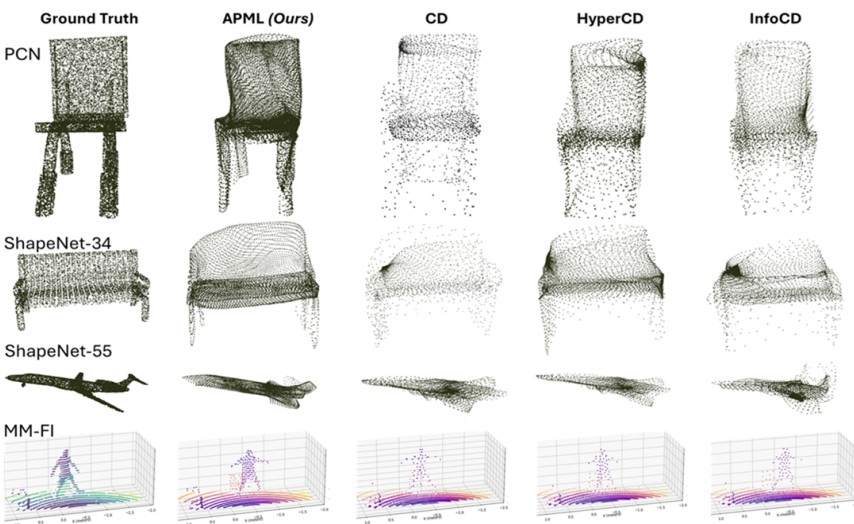

Figure 1: Impact of loss functions on point cloud completion (using FoldingNet), and generation (using CSI2PC). Compared with Chamfer distance-based losses, APML preserves structure better in sparse regions, reduces clumping, and generalizes across input modalities.

## Abstract

Training deep learning models for point cloud prediction tasks such as shape completion and generation depends critically on loss functions that measure discrepancies between predicted and ground-truth point sets. Commonly used functions such as Chamfer Distance (CD), HyperCD, InfoCD and Density-aware CD rely on nearest-neighbor assignments, which often induce many-to-one correspondences, leading to point congestion in dense regions and poor coverage in sparse regions. These losses also involve non-differentiable operations due to index selection, which may affect gradient-based optimization. Earth Mover Distance (EMD) enforces one-to-one correspondences and captures structural similarity more effectively, but its cubic computational complexity limits its practical use. We propose the Adaptive Probabilistic Matching Loss (APML), a fully differentiable approximation of one-to-one matching that leverages Sinkhorn iterations on a temperature-scaled similarity matrix derived from pairwise distances. We analytically compute the temperature to guarantee a minimum assignment probability, eliminating manual tuning. APML achieves near-quadratic runtime, comparable to Chamfer-based losses, and avoids non-differentiable operations. When integrated into state-of-the-art architectures (PoinTr, PCN, FoldingNet) on ShapeNet benchmarks and on a spatio-temporal Transformer (CSI2PC) that *generates* 3-D human point clouds from WiFi-CSI measurements, APM loss yields faster convergence, superior spatial distribution, especially in low-density regions, and improved or

on-par quantitative performance without additional hyperparameter search. The code is available at: `https://github.com/apm-loss/apml`.

# 1 Introduction

Point sets are a primary representation for three-dimensional data acquired through sensors such as LiDAR, structured-light scanners, and depth cameras [13]. They are widely used in geometric learning tasks, including surface reconstruction, object generation, shape completion, and registration across different views [9, 21, 31, 33], as shown in Figure 1. Many of these applications involve learning to map an input representation to a set of output points, where supervision requires aligning predicted points with a ground truth set. This alignment is typically enforced through a loss function that quantifies the similarity between point sets. Since these sets are unordered and may vary in cardinality or density, standard vector-based loss functions are not directly applicable. This has motivated the development of set-based loss functions that directly compare point sets during training [10, 1].

Widely adopted metrics such as Chamfer distance (CD) offer computational efficiency but often struggle with accurately capturing geometric details due to limitations such as sensitivity to outliers, tendency to cause point clustering, and issues arising from discrete nearest-neighbor assignments [1, 17]. Conversely, Earth Mover's Distance (EMD) provides superior geometric fidelity by encouraging one-to-one correspondences but is typically too computationally expensive for direct use in large-scale deep learning [22, 7]. Although several modifications to CD have been proposed to address some of its shortcomings [28, 17, 16], a fundamental need persists for a loss function that robustly approximates the desirable one-to-one matching properties of EMD without its prohibitive cost, while also offering smooth gradients suitable for training modern deep networks.

To address this gap, we introduce the **Adaptive Probabilistic Matching Loss (APML)**. APML is a novel, fully differentiable loss function that applies principles from optimal transport to establish soft, probabilistic correspondences between point sets. By approximating a transport plan through application of an efficient Sinkhorn-based mechanism, combined with a key innovation, a data-driven, analytically derived temperature schedule, APML is designed to overcome the limitations of prior methods. This adaptive temperature controls the sharpness of the probabilistic assignments without requiring manual regularization tuning, thereby contributing to stable training and encouraging the generation of high-fidelity point clouds with good structural coherence and surface coverage. Our approach is designed to be a broadly applicable tool for multiple point cloud prediction tasks.

The main contributions of this work are the following:

- We introduce **Adaptive Probabilistic Matching Loss (APML)**, a novel loss function which enforces *soft* one-to-one correspondences. This approach mitigates common shortcomings of Chamfer Distance, such as point clumping, density bias, and outlier sensitivity. APML approximates the matching quality of Earth Mover's Distance with near-quadratic complexity in the point count and a runtime comparable to CD-based losses.

- We introduce an adaptive temperature selection mechanism where each row and column in the transport plan assigns a minimum probability mass ($p_{min}$). This closed-form schedule removes the need for manual tuning of the Sinkhorn regularizer and adapts to the local geometric context of the point sets.

- We perform quantitative and qualitative evaluations on standard point cloud completion benchmarks, including ShapeNet and PCN, using three well-established backbone models. These evaluations show that APML obtains performance improvements or comparable results relative to established CD-based losses when measured by metrics sensitive to structural fidelity, such as EMD.

- We present real-world data evaluations on the challenging MM-Fi dataset. For these, a transformer-based architecture predicts 3D point clouds of humans in indoor spaces using WiFi Channel State Information (CSI) data as an input modality. These experiments demonstrate the ability of APML to improve training stability and the preservation of structural details and surface coverage, in a different task and domain.

APML is designed as a drop-in replacement for CD, requiring minimal changes in system implementation and introducing only one interpretable hyperparameter.

## 2 Related Work

**Point Cloud Prediction Tasks.** Point cloud prediction tasks include completion, generation, and reconstruction from partial or transformed inputs. These tasks are relevant to applications such as 3D scene understanding, autonomous systems, and human-centered sensing. Learning-based models have become the standard approach due to their ability to infer dense geometry from incomplete or abstract inputs [13].

A representative method for shape completion is PCN [33], which builds on PointNet [21] and FoldingNet [31] using an encoder–decoder design that refines a coarse output via grid deformation. This framework has been extended in models such as SnowflakeNet [29], which applies progressive refinement, and Transformer-based approaches like PoinTr [32] and SeedFormer [35], which model long-range dependencies through attention mechanisms. These models typically combine coarse-to-fine stages to first establish global shape and then improve local detail. Alongside completion from visual inputs, recent work has explored point cloud generation from RF-based signals such as WiFi-CSI. The CSI2PC model [18] applies a spatio-temporal Transformer to map CSI data into structured 3D point clouds, processing amplitude and phase information across antennas and subcarriers. It generates representations of indoor scenes with humans and furniture. Evaluation on the MM-Fi dataset [30] demonstrates generalization to unseen subjects and environments, supporting its relevance in joint communication and sensing [18]. These architectures highlight the need for effective supervision in learning from unordered point sets. Since predictions are sets without fixed ordering, loss functions must provide permutation-invariant comparisons with accurate geometric feedback. This motivates the development of distance metrics suitable for guiding such models during training.

**Point Cloud Distance Metrics and Loss Functions.** Supervising deep learning models for point cloud completion and generation requires loss functions that can compare unordered sets of points. Since point clouds may vary in density and cardinality, standard vector-based losses are not directly applicable. Instead, permutation-invariant set-based distances such as Chamfer Distance (CD) [10] and Earth Mover's Distance (EMD) [1] are commonly used. CD computes nearest-neighbor distances between the two sets in both directions and averages them. It is widely adopted due to its computational efficiency and ease of implementation in frameworks used for models like PCN [33], PoinTr [32], and CSI2PC [18]. However, CD allows many-to-one mappings, which often lead to clustering in dense regions and poor coverage in sparse areas. Its reliance on discrete assignments also introduces non-differentiability, affecting gradient-based optimization [1, 17]. EMD, or Wasserstein-1 distance, addresses these issues by computing a one-to-one correspondence that minimizes the total transport cost between sets [10, 22]. This makes it more effective in preserving global shape structure and assigning geometrically meaningful matches. Nonetheless, its cubic complexity in the number of points [4] and requirement for equal cardinality render it impractical for training deep models at scale [7].

To improve over CD while avoiding the cost of exact EMD, several modifications have been introduced. Density-aware Chamfer Distance (DCD) incorporates local density weights to balance sparse and dense regions [28], though it may still amplify the influence of isolated points. Hyperbolic Chamfer Distance (HyperCD) modifies the metric space to reduce the effect of distant mismatches and sharpen local gradient behavior [17]. Contrastive Chamfer Distance (InfoCD) incorporates a regularization term inspired by contrastive learning to spread predicted points across the target shape, improving coverage and robustness to sampling noise [16]. Another recently proposed approach is the Learnable Chamfer Distance (LCD) [14], which replaces static matching rules with dynamically predicted weight distributions. LCD introduces a learnable attention mechanism that adaptively emphasizes critical reconstruction errors, improving convergence and representation learning. Through adversarial training, LCD identifies structural defects and adjusts local weightings in the distance computation, resulting in faster convergence and enhanced geometric fidelity while maintaining the computational simplicity of CD. Another alternative formulation is the Sliced Wasserstein Distance (SWD) [19], which approximates the Wasserstein transport by projecting point clouds into lower-dimensional spaces and averaging transport costs across multiple random slices. SWD preserves the geometric consistency of EMD while having a computational complexity comparable to CD, making it suitable for evaluating or training models where EMD would be computationally infeasible. Despite their improvements, DCD, HyperCD, InfoCD, LCD, and SWD remain based on fixed-point correspondences or approximations of transport that may still lead to partial many-to-one mappings and suboptimal gradient flow in sparse or non-uniform distributions.

**Probabilistic Matching and Optimal Transport.** Optimal transport (OT) provides a formalism for measuring dissimilarity between distributions by computing the minimal cost of reassigning mass from one to another [20]. In the context of point cloud comparison, Earth Mover's Distance (EMD) is a special case of OT that seeks a cost-minimizing transport plan between two sets, subject to marginal constraints. EMD produces one-to-one matchings and captures global structure but requires solving a linear program with cubic time complexity [22], making it impractical for large-scale learning. It also assumes equal cardinality between sets, which is often not the case in real applications.

To address these limitations, Cuturi [7] introduced an entropy-regularized formulation of OT that smooths the transport objective by adding a negative entropy term. This modification enables a fast and differentiable approximation using the Sinkhorn-Knopp algorithm, which iteratively normalizes the rows and columns of a cost-derived matrix. The result is a doubly stochastic matrix that defines soft, probabilistic correspondences between point sets [7, 27]. Unlike hard nearest-neighbor matching, this approach improves gradient stability during training and allows for continuous optimization. The regularization parameter $\varepsilon$ controls the sharpness of the assignment, with lower values approaching hard matchings and higher values producing smoother distributions. This tunable behavior has made Sinkhorn-based OT a common tool in differentiable applications [12], including 3D point cloud registration and alignment under uncertainty [24, 23]. These formulations support learning in cases where exact matching is ambiguous or ill-defined. Building on this foundation, we propose a loss function that approximates one-to-one matching through soft, differentiable assignments while dynamically adjusting assignment sharpness using local distance structure.

## 3 Adaptive Probabilistic Matching Loss (APML)

Drawing from the principles of optimal transport and the computational efficiency of entropy-regularized approaches such as the Sinkhorn algorithm [7], we introduce the *Adaptive Probabilistic Matching Loss (APML)*. APML is a fully differentiable loss function that compares unordered point sets by constructing a soft, probabilistic approximation of one-to-one correspondences. The objective is to provide the geometric supervision properties of transport-based losses while avoiding the computational burden and set cardinality constraints associated with exact methods. Unlike nearest-neighbor-based losses, such as CD and its variants, APML does not rely on discrete index selection, which can interfere with gradient propagation. A key distinction from existing Sinkhorn-based approaches is the introduction of a data-dependent mechanism that adaptively selects the temperature parameter controlling the sharpness of the transport distribution. This parameter is computed analytically from the pairwise distances, ensuring that each point maintains a minimum level of probabilistic assignment and eliminating the need for manual tuning of the regularization.

The APML procedure begins by constructing a soft assignment matrix from the pairwise distances between predicted and ground truth point sets. Rather than computing hard matchings, the assignment distributes mass across all candidates based on temperature-scaled similarities. Assignments are computed independently in both directions and averaged to maintain consistency. The resulting matrix is then refined using normalization steps to approximate a doubly stochastic transport plan. The adaptive control of sharpness allows the loss to adjust locally to the geometry of each pairwise comparison, providing more stable gradients and improved point coverage.

Before defining the complete loss function, we describe the mathematical preliminaries. Let $\hat{X} \in \mathbb{R}^{B \times N \times d}$ denote the predicted point sets and $X \in \mathbb{R}^{B \times M \times d}$ the ground truth point sets, where $B$ is the batch size, $N$ and $M$ are the number of predicted and ground truth points, respectively, and $d$ is the spatial dimensionality. For each batch element $b \in \{1, \ldots, B\}$, the pairwise cost matrix $C_b \in \mathbb{R}^{N \times M}$ is computed using the Euclidean distance:

$$C_{b,i,j} = \left\| \hat{X}_{b,i} - X_{b,j} \right\|_2 , \tag{1}$$

where $i \in \{1, \ldots, N\}$ indexes the predicted points and $j \in \{1, \ldots, M\}$ indexes the ground truth points. The construction of the transport matrix, the adaptive temperature schedule, and the final loss objective are defined in the following subsections.

**Adaptive Softmax.** To generate soft correspondences between point sets, the APML method defines an adaptive softmax function. This function is designed to ensure that for any given point, its resulting probability distribution over potential matches assigns at least a minimum probability,

$p_{\min} \in (0, 1)$, to its most likely match (i.e., the match with the lowest cost). This mechanism is applied independently to each row and each column of the cost matrix $C$.

Consider a generic cost vector $\mathbf{c} = (c_1, c_2, \ldots, c_K) \in \mathbb{R}^K$, representing the costs from one point to $K$ other points. The adaptive softmax computation for this vector proceeds as follows:

1. **Cost Normalization:** The cost vector $\mathbf{c}$ is first normalized by subtracting its minimum value to prevent potential numerical issues with large cost values in the exponential function and to focus on relative differences. Let $\tilde{\mathbf{c}}$ be the normalized cost vector, defined as $\tilde{c}_j = c_j - \min_{l=1,\ldots,K} c_l$ for $j = 1, \ldots, K$. Thus, $\min_j \tilde{c}_j = 0$.

2. **Local Gap Definition:** Let $\tilde{c}_{(1)}$ and $\tilde{c}_{(2)}$ be the smallest (i.e., 0) and the second smallest different values in $\tilde{\mathbf{c}}$, respectively. If all elements are identical (i.e., $\tilde{c}_j = 0$ for all $j$), $\tilde{c}_{(2)}$ can be considered notionally large or handled as a special case (see step 4). The local gap, $g$, is defined to ensure a margin if $\tilde{c}_{(2)}$ is very close to $\tilde{c}_{(1)}$ as $g = \tilde{c}_{(2)} + \delta$, where $\delta > 0$ is a small positive constant (e.g., $10^{-6}$) added for numerical stability, particularly if $\tilde{c}_{(2)} = 0$.

3. **Adaptive Temperature Calculation:** To ensure that the probability assigned to the element with the minimum cost (i.e., $\tilde{c}_{(1)} = 0$) is at least $p_{\min}$, we solve the following inequality for the temperature $T > 0$, assuming $K > 1$:

$$\frac{\exp(-T \cdot 0)}{\exp(-T \cdot 0) + \sum_{k=2}^{K} \exp(-T\tilde{c}_{(k)})} \approx \frac{1}{1 + (K-1)\exp(-Tg)} \geq p_{\min}. \tag{2}$$

The approximation uses the second smallest cost $\tilde{c}_{(2)}$ (via $g$) as a representative for other non-minimal costs to simplify the derivation of $T$. This leads to the adaptive temperature:

$$T = \frac{-\log\left(\frac{1-p_{\min}}{(K-1)p_{\min}}\right)}{g}. \tag{3}$$

This expression for $T$ is valid under the conditions $K > 1$ and $0 < p_{\min} < 1$, which ensure that the logarithmic term is well-defined and strictly positive. When $K = 1$, the assignment is trivially deterministic with probability 1, and no temperature scaling is required. The constraints $(K-1)p_{\min} > 0$ and $1 - p_{\min} > 0$ must hold to avoid numerical instability and to ensure that the denominator within the logarithm remains positive.

4. **Numerical Stability for Multiple Minima:** If multiple elements in the cost vector $\mathbf{c}$ share the same minimum value (i.e., after normalization, $\tilde{c}_{(1)} = \tilde{c}_{(2)} = 0$), the gap $g$ becomes approximately equal to $\delta$. A small gap leads to a large temperature $T$, which may produce numerically unstable behavior and overly concentrated assignments. To prevent this, if $\tilde{c}_{(2)} < \epsilon_g$ for a small threshold $\epsilon_g$ (e.g., $10^{-5}$), we override the temperature-scaled softmax with a uniform probability distribution. Let $P_j$ denote the assignment probability to the $j$-th element of the vector, where then the assignment is defined as:

$$P_j = \frac{1}{K}, \quad \text{for all } j = 1, \ldots, K, \tag{4}$$

where $K$ is the number of elements in $\mathbf{c}$. This guarantees numerical stability in cases where multiple effective minima are present and avoids assigning excessively high confidence to any individual element.

5. **Scaled Softmax Application:** If the uniform override (Step 4) is not triggered, the temperature $T$ from Step 3 is used to compute the soft probability distribution $\mathbf{P} = (P_1, \ldots, P_K)$ over the $K$ elements:

$$P_j = \frac{\exp(-T\tilde{c}_j)}{\sum_{k=1}^{K} \exp(-T\tilde{c}_k)}. \tag{5}$$

The adaptive softmax procedure detailed above (Steps 1-5) is then applied to the overall cost matrix $C_b \in \mathbb{R}^{N \times M}$ (for each batch element $b$; subscript $b$ is omitted below for simplicity) to generate two initial probability matrices, $P_1$ and $P_2$:

- The matrix $P_1 \in \mathbb{R}^{N \times M}$ is obtained by applying the 5-step adaptive softmax procedure row-wise to $C$. For each row $i$ of $C$ (i.e., for the $i$-th predicted point $\hat{X}_i$), the input cost vector is $\mathbf{c}_{i,\cdot} = (C_{i,1}, \ldots, C_{i,M}) \in \mathbb{R}^M$. In this application, $K = M$. Each row of $P_1$ thus forms a probability distribution: $\sum_{j=1}^{M} (P_1)_{ij} = 1$ for each $i = 1, \ldots, N$.

- The matrix $P_2 \in \mathbb{R}^{N \times M}$ is obtained by applying the 5-step adaptive softmax procedure column-wise to $C$. For each column $j$ of $C$ (i.e., for the $j$-th ground truth point $X_j$), the input cost vector is $\mathbf{c}_{\cdot,j} = (C_{1,j}, \ldots, C_{N,j}) \in \mathbb{R}^N$. In this application, $K = N$. Each column of $P_2$ forms a probability distribution: $\sum_{i=1}^{N} (P_2)_{ij} = 1$ for each $j = 1, \ldots, M$.

To enforce consistency between these two directional perspectives (predicted-to-ground truth and ground truth-to-predicted), the resulting probability matrices $P_1$ and $P_2$ are averaged element-wise:

$$P = \frac{1}{2}(P_1 + P_2). \tag{6}$$

This matrix $P \in \mathbb{R}^{N \times M}$ represents the initial symmetrized soft assignment probabilities that will be further refined by Sinkhorn normalization.

**Sinkhorn Normalization.** The symmetrized probability matrix $P \in \mathbb{R}^{N \times M}$, obtained from the adaptive softmax stage (Equation (6)), represents initial soft correspondences. However, this matrix $P$ is not guaranteed to be doubly stochastic; that is, its row sums and column sums may not consistently adhere to the marginal constraints of a transport plan (e.g., rows summing to $1/N$ and columns to $1/M$ for uniform marginals, or more generally, rows and columns summing to 1 if $P$ is to be interpreted as a joint probability distribution between individual points).

To refine $P$ into an approximate doubly stochastic matrix, which better reflects a coherent transport plan, we apply Sinkhorn-Knopp normalization [25]. This is an iterative algorithm that alternates between normalizing the rows and columns of the matrix to sum to specific values (typically 1 in this context for each row and column, assuming we want $P_{ij}$ to represent the probability of matching point $\hat{X}_i$ to $X_j$ such that each point is fully "assigned"). The iterative process is performed for a fixed number of iterations, denoted as $L_{\text{iter}}$ (e.g., $L_{\text{iter}} = 20$). In each iteration $l = 1, \ldots, L_{\text{iter}}$, the following two normalization steps are applied sequentially to the matrix $P$ (denoting the matrix at the beginning of an iteration step as $P$ and its updated version also as $P$ for simplicity):

1. **Column Normalization Step:** Each element $P_{ij}$ is divided by the sum of its respective column. For all $i = 1, \ldots, N$ and $j = 1, \ldots, M$:

$$P_{ij} \leftarrow \frac{P_{ij}}{\sum_{k=1}^{N} P_{kj} + \varepsilon_{\text{stab}}}, \tag{7}$$

This step ensures that after its application, each column of $P$ sums approximately to 1 (i.e., $\sum_{i=1}^{N} P_{ij} \approx 1$ for each $j$).

2. **Row Normalization Step:** Each element $P_{ij}$ is then divided by the sum of its respective row. For all $i = 1, \ldots, N$ and $j = 1, \ldots, M$:

$$P_{ij} \leftarrow \frac{P_{ij}}{\sum_{k=1}^{M} P_{ik} + \varepsilon_{\text{stab}}}. \tag{8}$$

This step ensures that after its application, each row of $P$ sums approximately to 1 (i.e., $\sum_{j=1}^{M} P_{ij} \approx 1$ for each $i$).

In these equations, $\varepsilon_{\text{stab}}$ is a small positive constant (e.g., $10^{-8}$) added to the denominator to prevent division by zero, ensuring numerical stability, particularly if some row or column sums happen to be zero or very close to zero during the iterations. After $L_{\text{iter}}$ iterations of these alternating normalizations, the resulting matrix $P \in \mathbb{R}^{N \times M}$ serves as the refined, approximately doubly stochastic transport plan representing the soft correspondences between the predicted point set $\hat{X}$ and the ground truth point set $X$.

**APML Objective Function.** Having computed the refined, approximately doubly stochastic transport plan $P_b \in \mathbb{R}^{N \times M}$ for each batch element $b$ (as detailed in the Sinkhorn Normalization subsection,

using the output of Equation (7) or (8) after $L_{\text{iter}}$ iterations), and utilizing the original pairwise cost matrix $C_b \in \mathbb{R}^{N \times M}$, the Adaptive Probabilistic Matching Loss ($\mathcal{L}_{\text{APML}}$) is defined. The loss $\mathcal{L}_{\text{APML}}$ is computed as the expected matching cost under the learned soft assignment probabilities $P_{b,i,j}$. This is averaged over all predicted points, all ground truth points (implicitly through the sum over $j$ weighted by $P_{b,i,j}$ which itself sums to 1 over $j$ for each $i$), and all elements in the batch $B$:

$$\mathcal{L}_{\text{APML}} = \frac{1}{B} \sum_{b=1}^{B} \left( \sum_{i=1}^{N} \sum_{j=1}^{M} P_{b,i,j} \cdot C_{b,i,j} \right). \tag{9}$$

In this formulation, $P_{b,i,j}$ represents the refined probability of matching the $i$-th predicted point to the $j$-th ground truth point for the $b$-th element in the batch, and $C_{b,i,j}$ is the corresponding cost (distance) between them. The inner double summation $\sum_{i=1}^{N} \sum_{j=1}^{M} P_{b,i,j} \cdot C_{b,i,j}$ can be interpreted as the Frobenius inner product $\langle P_b, C_b \rangle_F$, representing the total cost for the $b$-th pair of point sets under the soft assignment $P_b$.

This loss formulation encourages accurate point-to-point alignment by penalizing mismatches according to the learned soft correspondences, while the differentiability of the transport plan $P$ (due to the differentiable nature of the adaptive softmax and Sinkhorn steps) ensures a smooth gradient flow for optimization. The adaptive control of assignment sharpness and the enforcement of bidirectional consistency, as detailed in the preceding subsections, contribute to APML providing a robust and efficient mechanism for soft matching in learning-based geometric tasks.

## 4  Experimental Evaluation

**Experimental Setup.** All experiments are conducted using Python 3.11, PyTorch 2.5, and CUDA 12.8. The models are trained on a publicly available supercomputer, using 4 nodes, each equipped with four Nvidia Volta V100 GPUs with 32 GB of memory each and 2 Intel Xeon processors *Cascade Lake*, with 20 cores each running at 2,1 GHz. We use the official open-source implementations of FoldingNet, PCN, and PoinTr from [34], with their default training configurations, including learning rate, optimizer, and batch size. For point cloud generation from WiFi Channel State Information, we use the CSI2PointCloud model [18], a transformer-based architecture that estimates 3D spatio-temporal point clouds from raw CSI input. The implementation is available online [2] and is used with its default training protocol. We use the official implementations of CD, HyperCD, and InfoCD for loss computation. For our proposed APML, the hyperparameters were set consistently across most experiments, unless otherwise noted: minimum assignment probability $p_{\text{min}} = 0.8$, adaptive softmax gap margin $\delta = 10^{-6}$, gap threshold for uniform override $\epsilon_g = 10^{-5}$, number of Sinkhorn iterations $L_{\text{iter}} = 10$, and Sinkhorn stability constant $\varepsilon_{\text{stab}} = 10^{-8}$. Any deviations from these settings for specific experiments will be explicitly mentioned. For all experiments, we use a fixed threshold of $\tau = 0.01$ when computing the F1-score. This value corresponds to the default setting in the PoinTr evaluation framework and is commonly adopted in the literature for point cloud completion tasks, as it provides a reasonable balance between spatial tolerance and sensitivity to local geometric accuracy.

**Evaluation Benchmark.** We evaluate APML against three leading point cloud supervision objectives, CD, InfoCD, and HyperCD, across a diverse set of benchmarks encompassing both point cloud completion and cross-modal generation. Our experiments span three datasets: PCN [33], ShapeNet (SN34/SN55) [6], and MM-Fi [30], covering both synthetic and real-world modalities, including the challenging task of generating 3D point clouds from WiFi CSI signals. In addition, we include the synthetic PCOU3D dataset [5], as shown in Figure 3, designed as a controlled environment to systematically evaluate stability and density robustness under five defined input regimes. These datasets enable evaluation in both single-category and multi-category settings. Performance is assessed using standard metrics: Chamfer Distance [10], Earth Mover's Distance [22], and F1 score [33] at a fixed threshold. Detailed descriptions of the datasets and metric definitions are provided in Appendix B.1 and Appendix B.2. In addition, we include two widely used alternative loss formulations in the comparison: the density-aware Chamfer variant DCD, which enhances CD by accounting for local point density, and SWD, an EMD-inspired distance based on sliced Wasserstein projections. We report DCD and SWD on configurations where they are applicable (for example, SWD requires equal cardinality between prediction and ground truth) and where we conducted additional runs; other entries are left unreported.

**Experimental Results.** Table 1 summarizes our main validation results. For each completion dataset and backbone, we report the F1 score and EMD×100; for generation, we report CD and EMD×100. In all settings, APML consistently achieves significantly lower EMD compared to previous losses, often by wide margins of 15–81%, while maintaining comparable or slightly better F1 scores. This trend holds for both simpler models like FoldingNet and more expressive architectures like PoinTr. Where available, the DCD and SWD rows show that APML reduces EMD×100 relative to these surrogates while matching or improving F1 (e.g., SN34 with PoinTr and FoldingNet), and on MM-Fi achieves substantially lower EMD×100 than DCD with comparable CD. The disaggregated results, additional metrics (CD L1 & CD L2) for the PCN dataset, and an analysis of statistical significance are reported in Appendix C.

Table 1: Aggregated validation results with additional surrogates (DCD, SWD). Completion uses F1 (↑) and EMD×100 (↓); generation uses CD (↓) / EMD×100 (↓). A dash (–) denotes not reported in our extended runs (e.g., due to applicability under our setup, such as SWD equal-cardinality assumption or unstable training). **Bold** denotes the best result within each row.

| Dataset / Backbone | CD [10] | HCD [17] | InfoCD [16] | DCD [28] | SWD [19] | **APML** |
|---|---|---|---|---|---|---|
| | | | Loss function | | | |
| | Point cloud completion (F1 / EMD×100) | | | | | |
| **PCN** | | | | | | |
|   PCN | 0.60 / 12.87 | **0.64** / 6.53 | 0.64 / 5.54 | – | – | 0.62 / **4.72** |
|   FoldingNet | 0.43 / 28.71 | 0.52 / 22.93 | 0.54 / 21.24 | – | – | **0.56 / 5.34** |
|   PoinTr | 0.75 / 9.46 | **0.77** / 8.79 | 0.43 / 9.72 | – | – | 0.67 / **5.62** |
| **ShapeNet-55** | | | | | | |
|   FoldingNet | 0.11 / 26.50 | 0.15 / 23.20 | 0.17 / 19.55 | – | – | **0.20 / 9.07** |
|   PoinTr | 0.46 / 7.55 | **0.56** / 9.02 | 0.35 / 10.35 | – | – | 0.51 / **6.08** |
| **ShapeNet-34** | | | | | | |
|   FoldingNet | 0.11 / 27.71 | 0.19 / 16.53 | 0.18 / 22.26 | 0.19 / 11.52 | 0.11 / 11.98 | **0.20 / 9.49** |
|   PoinTr | 0.42 / 11.88 | **0.52** / 12.03 | 0.35 / 13.41 | 0.47 / 9.48 | – | 0.50 / **8.14** |
| **SN Unseen-21** | | | | | | |
|   FoldingNet | 0.11 / 32.10 | 0.19 / 19.27 | 0.18 / 25.47 | – | – | **0.20 / 10.42** |
|   PoinTr | 0.42 / 12.89 | **0.52** / 13.27 | 0.35 / 15.19 | – | – | 0.49 / **9.19** |
| | Point cloud generation from Wi-Fi (CD / EMD×100) | | | | | |
| **MM-Fi** | | | | | | |
|   CSI2PC | 0.150 / 34.20 | 0.149 / 36.75 | **0.147** / 35.19 | 0.148 / 25.68 | – | 0.152 / **14.11** |

Interestingly, while F1 reflects discrete matching accuracy, it does not fully capture perceptual alignment or structure preservation. In cases where F1 scores are similar, qualitative samples (see Fig. 1) and EMD values demonstrate that APML produces more geometrically faithful reconstructions. This suggests that APML acts as a regularizer toward semantically meaningful one-to-one alignments that Chamfer-based objectives often miss. On SN34, APML lowers EMD×100 with respect to DCD while matching or improving F1 (FoldingNet and PoinTr). In MM-Fi, APML achieves a CD comparable to DCD but reduces EMD×100 by a large margin (from 25.68 to 14.11). A comparative evaluation of visual perception against the metrics for point clouds can be seen in Appendix D.1.

**Stability Analysis**. We evaluated APML robustness to input density and batch composition variations using the synthetic PCOU3D dataset (four primitives, five regimes A–E: sparse, dense, mixed, completion). We trained FoldingNet comparing APML with CD and InfoCD. The decoder was always supervised with the 1024-point ground truth. APML was the most stable in all regimes, with $\sigma_{\text{CD}} = 0.0013$ and $\sigma_{\text{EMD}} = 0.0025$, an order of magnitude lower than CD or InfoCD. On sparse inputs (regime A) CD's EMD increased 17% due to many-to-one point collapses, while APML changed $< 6\%$, near one-to-one correspondences. This confirms that the adaptive temperature mechanism effectively regularizes matching, reducing sensitivity to density and batch composition without introducing instability. Further details are provided in Appendix C.1.

**Sensitivity to $p_{\text{min}}$.** APML's only interpretable hyperparameter $p_{\text{min}}$ places a lower bound on the probability mass for the closest target in the adaptive softmax. We assessed its impact on PCOU3D and MM-Fi, varying $p_{\text{min}} \in \{0.01, 0.1, 0.5, 0.8\}$. Across sparse, dense, and completion regimes, CD and EMD vary $< 5\%$ for $p_{\text{min}} \in [0.01, 0.8]$. This weak dependence stems from the analytic temperature $T$ in Eq. 3, which scales inversely with the local distance gap $g$. Small gaps increase $T$, distributing probability; large gaps decrease $T$, yielding sharp assignments. Because $T$ is recomputed row by row, changing $p_{\text{min}}$ within this broad band leaves the transport plan and its gradients nearly unchanged. Quantitative results appear in Table 6 and Appendix C.2.

**Ablation of APML components.** We isolated the role of each component using PCOU3D regimes A, B, and E. Removing Sinkhorn normalization markedly increases EMD, especially in dense and completion settings, due to unconstrained marginals. Using a fixed temperature without adaptivity ($T = 0.1$) leads to unstable optimization and much higher losses, indicating the adaptive link between $T$ and $g$ is necessary. Dropping bidirectional symmetrization introduces directional bias and severely degrades completion. The complete APML configuration yields the lowest CD and EMD across the three regimes. Quantitative results appear in Table 7 and Appendix C.2.

Next, we analyze the convergence behavior and runtime characteristics. Table 2 compare wall-clock training time and memory usage for FoldingNet on the PCN dataset using four different loss functions. Although APML introduces a $\sim$30% increase in training time per epoch relative to CD, its convergence plot, shown in Figure 2, shows substantially faster improvement in validation F1, reaching strong performance much earlier in training. In practice, this means that APML could achieve competitive results in significantly fewer epochs, despite all models here being trained for 150 epochs for consistency. Other backbones show similar relative overhead ($\sim$15% and 4-5 x RAM for FoldingNet).

Table 2: Wall-clock training time and peak GPU memory usage for FoldingNet trained on ShapeNet-55 (150 epochs on V100 32GB, batch size 128).

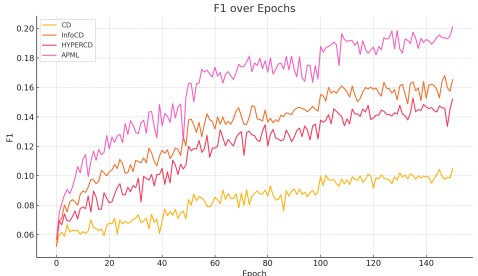

Figure 2: F1 score on SN55 validation set across 150 epochs with FoldingNet. APML converges faster and achieves higher final performance.

| Loss | Time | Mem |
|---|---|---|
| CD | 55 h | < 64 GB |
| HyperCD | 57 h | < 64 GB |
| InfoCD | 58 h | < 64 GB |
| **APML** | 76 h | < 320 GB |

Although APML's memory usage is higher due to the quadratic cost of maintaining a pairwise cost matrix, this matrix is consistently sparse with over 90% of entries falling near zero before Sinkhorn normalization. Leveraging sparsity, we can prune *almost* 99.9% of elements of value $< 10^{-8}$ with negligible impact on both qualitative and quantitative results, as seen in Appendix D.3).

## 5  Discussion

Deep point-cloud supervision has long been torn between *efficiency* (Chamfer Distance objectives) and *geometric fidelity* (exact Earth Mover's Distance, EMD). Our results show that the proposed APML largely closes this gap: at a computational cost from $\sim$15 to 30% higher than CD loss, APML achieves enough one-to-one alignments to reduce EMD by $\sim$15–80% across three architectures and three datasets. In addition, APML reaches its peak validation EMD in fewer epochs than Chamfer-based training, further lowering the effective computational budget.

Exact EMD is a useful reference for correspondence quality, but its cubic complexity and equal-cardinality constraint make full-scale training and evaluation impractical for high-resolution completion and cross-modal generation. We therefore include comparisons to DCD, a density-aware CD variant, and SWD, an EMD-inspired sliced Wasserstein formulation. In the settings where these methods are applicable and stable, APML reduces EMD$\times$100 and improves qualitative alignment (Table 1). Full EMD supervision at scale is out of scope, and the appendix provides added qualitative comparisons and notes the equal-cardinality and stability constraints that shaped this evaluation.

The controlled study on PCOU3D confirms robustness to input density and batch composition: across regimes A–D the standard deviation of APML is $\sigma_{\text{CD}} = 0.0013$ and $\sigma_{\text{EMD}} = 0.0025$, well below CD and InfoCD, and sparse-only training increases EMD by about 17% for CD but by less than 6% for APML. The analytic temperature that depends on the local distance gap maintains soft assignments when targets are ambiguous and sharp assignments when a clear match exists, which stabilizes gradients without manual tuning.

These improvements are enabled by a combination of design choices in APML. First, the data-driven temperature scheme eliminates the need for a manually tuned Sinkhorn regulariser, as used in

related optimal transport surrogates [7], yet remains numerically stable; rows with duplicate minima occurred in fewer than 0.6% of batches. Surprisingly, APML sometimes *increases* CD-L1/L2 on the strongest backbone (PoinTr), while still improving perceptual metrics and visual quality, as illustrated in Appendix A. This divergence reinforces recent critiques that Chamfer distance overweights dense regions and correlates poorly with human judgment [17]. The component ablations further indicate that temperature and Sinkhorn are coupled parts of a single probabilistic matching process. Fixing the temperature or removing Sinkhorn sharply degrades performance, and dropping the bidirectional symmetrization introduces directional bias, especially in completion. The full configuration consistently yields the lowest CD and EMD on PCOU3D. Second, the benefit is architecture-agnostic: even the weakest encoder (FoldingNet) enjoys a fivefold reduction in EMD, suggesting that APML also acts as a geometric regulariser when model capacity is limited. These gains persist without hyperparameter re-tuning when transferring from ShapeNet to the WiFi-CSI MM-Fi benchmark, underscoring the robustness of the adaptive temperature mechanism.

**Limitations.** Despite the strong empirical performance of APML, its general applicability across architectures, and the absence of extensive tuning requirements, certain limitations remain. First, although APML eliminates the Sinkhorn regulariser $\varepsilon$, it introduces a single hyper-parameter, the soft-assignment threshold $p_{\min}$. We hold $p_{\min} = 0.01$ constant for all experiments; no tuning was attempted. During training, the *resulting* maximum row/column probability typically increases to $\sim 0.8$, but this value is a *outcome*, not a preset. Extremely small or large $p_{\min}$ values can, in principle, destabilize training, and investigating this sensitivity remains a future work. Our ablation shows that CD and EMD vary by less than 5% once $p_{\min} \in [0.01, 0.8]$ on PCOU3D and MM-Fi, which is consistent with adaptive temperature computed from the local gap governing the effective sharpness.

Second, the required memory still scales quadratically with point count; for 16k points, the cost matrix consumes approximately 1.8 GB, which restricts the use of larger batch sizes. Empirically, most of the similarity values fall to near-zero values after the exponential transformation, rendering the matrix effectively sparse. We currently store it densely for simplicity; exploiting sparsity or low-rank factoring [3, 15, 26] could push the effective cost toward $O(N \log N)$ and is an obvious next step. Likewise, our runtime figures use plain PyTorch tensor ops; a fused CUDA kernel could further narrow the overhead when compared with CD.

Finally, our evaluation spans two synthetic completion sets and one real generation set. We have not yet measured *completion* in real-scan datasets such as ScanNet [8] or KITTI [11], nor generation beyond silhouettes. Thin structures and sensor noise may expose additional failure modes that affect all transport-based losses, although given the current empirical evidence, we find this unlikely.

**Future Work and Broader Impact.** Future work will explore learnable or schedule-based alternatives to $p_{\min}$, low-rank or sliced Sinkhorn variants to reduce memory usage, and a fully optimized CUDA implementation of APML. Extending evaluation to noisy, real-world scans and non-Euclidean domains (e.g., surfaces or graphs) is also a priority. These steps aim to bring APML closer to practical deployment in robotics, AR, and digital twin and simulation settings where perceptual structure is more important than point-wise precision. In contrast, the same technology might reduce the barrier to improve indoor sensing from commodity WiFi devices. APML's one-to-one regularization could enable lighter, energy-efficient models for edge devices, facilitating real-time 3-D feedback for low-vision navigation or affordable home robotics. At the same time, stronger completion and WiFi-based reconstruction lower the technical barrier for covert sensing and would require usage-restricted licenses and automated filters that reject models fine-tuned on non-consensual data.

# 6 Conclusion

We introduced **Adaptive Probabilistic Matching Loss (APML)**, a differentiable, near-quadratic surrogate for Earth Mover's Distance that brings one-to-one point-set alignment to deep point-cloud learning with only a modest ($\sim 15\%$) runtime overhead relative to Chamfer Distance. APML's analytically derived, data-driven temperature removes the need for manual Sinkhorn tuning, remains numerically stable across diverse inputs, and improves perceptual metrics, reducing EMD by 15–81% on three architectures and 24 ShapeNet classes, while maintaining strong generalisation to domain-specific WiFi-CSI data. The method's architecture-agnostic gains, minimal hyperparameter burden, and competitive efficiency position APML as a drop-in replacement for Chamfer-style losses in point cloud reconstruction, completion, and generation pipelines.

## Acknowledgments and Disclosure of Funding

This research was supported by the Business Finland 6G-WISECOM project (Grant 3630/31/2024), the University of Oulu, and the Research Council of Finland (formerly Academy of Finland) through the 6G Flagship Programme (Grant 346208). Sasan Sharifipour acknowledges the funding from the Finnish Doctoral Program Network on Artificial Intelligence, AI-DOC (decision number VN/3137/2024-OKM-6), supported by Finland's Ministry of Education and Culture and hosted by the Finnish Center for Artificial Intelligence (FCAI). The authors declare no conflict of interest.

Computational resources were provided by CSC – IT Center for Science, Finland, with model training performed on the Mahti and Puhti supercomputers.

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

# A   Technical Appendix: Algorithm description and theoretical analysis

This appendix collects algorithm pseudocode, and a theoretical analysis and comparison against other loss functions.

## A.1   APML Algorithm Summary

The computation of the Adaptive Probabilistic Matching Loss for a batch of predicted point sets $\hat{X}$ and ground truth point sets $X$ is summarized in Algorithm 1. This procedure includes pairwise cost computation, adaptive softmax with bidirectional matching, Sinkhorn normalization, and final loss evaluation.

**Input:** Predicted point sets $\hat{X} \in \mathbb{R}^{B \times N \times d}$, Ground truth point sets $X \in \mathbb{R}^{B \times M \times d}$
**Input:** Hyperparameters: $p_{\min}, \delta, \epsilon_g$ (adaptive softmax); $L_{\mathrm{iter}}, \varepsilon_{\mathrm{stab}}$ (Sinkhorn)
**Output:** Loss value $\mathcal{L}_{\mathrm{APML}}$
Initialize total loss: $\mathcal{L}_{\mathrm{total}} \leftarrow 0$;
**for** $b \leftarrow 1$ **to** $B$ **do**
  $\hat{X}_b \leftarrow \hat{X}[b,:,:]$, $X_b \leftarrow X[b,:,:]$;
  Compute cost matrix $C_b \in \mathbb{R}^{N \times M}$ where $(C_b)_{ij} = \|\hat{X}_{b,i} - X_{b,j}\|_2$;
  // Compute soft assignments from predicted to ground truth
  **for** $i \leftarrow 1$ **to** $N$ **do**
    $\mathbf{c}_{i,\cdot} \leftarrow (C_b)_{i,:}$;
    $(P_{1,b})_{i,:} \leftarrow \mathrm{AdaptiveSoftmaxVec}(\mathbf{c}_{i,\cdot}, M, p_{\min}, \delta, \epsilon_g)$;
  **end**
  // Compute soft assignments from ground truth to predicted
  **for** $j \leftarrow 1$ **to** $M$ **do**
    $\mathbf{c}_{\cdot,j} \leftarrow (C_b)_{:,j}$;
    $(P_{2,b})_{:,j} \leftarrow \mathrm{AdaptiveSoftmaxVec}(\mathbf{c}_{\cdot,j}, N, p_{\min}, \delta, \epsilon_g)$;
  **end**
  // Symmetrize
  $P_{\mathrm{init}} \leftarrow \frac{1}{2}(P_{1,b} + P_{2,b})$;
  // Apply Sinkhorn normalization
  $P \leftarrow P_{\mathrm{init}}$;
  **for** $l \leftarrow 1$ **to** $L_{iter}$ **do**
    **for** $j \leftarrow 1$ **to** $M$ **do**
      $P_{:,j} \leftarrow P_{:,j}/\left(\sum_{k=1}^{N} P_{k,j} + \varepsilon_{\mathrm{stab}}\right)$;
    **end**
    **for** $i \leftarrow 1$ **to** $N$ **do**
      $P_{i,:} \leftarrow P_{i,:}/\left(\sum_{k=1}^{M} P_{i,k} + \varepsilon_{\mathrm{stab}}\right)$;
    **end**
  **end**
  // Compute loss for current batch item
  $\mathcal{L}_b \leftarrow \sum_{i=1}^{N} \sum_{j=1}^{M} P_{ij} \cdot C_{b,ij}$;
  $\mathcal{L}_{\mathrm{total}} \leftarrow \mathcal{L}_{\mathrm{total}} + \mathcal{L}_b$;
**end**
$\mathcal{L}_{\mathrm{APML}} \leftarrow \mathcal{L}_{\mathrm{total}}/B$;
**return** $\mathcal{L}_{APML}$;

**Algorithm 1:** Adaptive Probabilistic Matching Loss (APML)

## A.2   Theoretical Analysis and Comparison with Other Loss Functions for Point Cloud Tasks

We provide a theoretical comparison of APML with representative loss functions commonly used in point cloud prediction tasks. The comparison focuses on computational complexity, sensitivity to outliers and local density variations, and the nature of the assignment strategy. The proposed APML is contrasted with methods based on nearest-neighbor correspondence and with loss functions inspired by optimal transport. The comparison is summarized in Table 3.

Unlike nearest-neighbor-based losses, which rely on discrete assignments and may suffer from clustering artifacts or instability in sparse regions, APML constructs a probabilistic transport plan that is refined through iterative normalization. Its formulation avoids the high complexity of exact optimal transport by using a fixed number of Sinkhorn scaling steps, while introducing an adaptive temperature mechanism that automatically adjusts the sharpness of the assignment based on the local cost structure. This allows APML to improve alignment quality without requiring manual regularization tuning.

Table 3: Theoretical comparison of point cloud loss functions. Complexity assumes $N$ predicted points, $M$ ground truth points, dimensionality $d$, and $L$ Sinkhorn iterations (for APML). For EMD, $K = \max(N, M)$ (complexity often cited assuming $N \approx M = K$).

| Loss Function | Core Mechanism | Complexity (Approx.) | Outlier Sensitivity | Density Sensitivity | Mapping Preference | Theoretical Advantages | Limitations |
|---|---|---|---|---|---|---|---|
| CD [10] | Nearest Neighbor (NN) | $O(NMd)$ (naive) or $O((N+M)\log M \cdot d)$ (with spatial structures) | High | Low | Many-to-one | Simplicity; Relatively Fast | Outlier & density issues; Clumping; Gradient quality |
| EMD [22] | Optimal Transport (Exact LP) | $O(K^3 \log K)$ | Low | High | One-to-one | High geometric fidelity; Robustness | Very high computational cost; Cardinality constraint (std. form) |
| DCD [28] | Density-weighted NN | $O(NMd)$ | Medium | Medium-High | Many-to-one | Attempts improved density awareness | Can boost sparse outliers; Weight tuning |
| HyperCD [17] | Hyperbolic space NN | $O(NMd)$ | Medium | Low | Many-to-one | Down-weights distant pairs (outlier robustness) | Requires $\alpha$ tuning; Still NN-based |
| InfoCD [16] | Contrastive NN Regularization | $O(NMd)$ + Contrastive Overhead | Medium | Medium (via spreading) | Many-to-one (encourages spreading) | Mitigates clumping; Improves coverage | Added setup complexity; Contrastive tuning; Still NN-based |
| **APML (ours)** | Sinkhorn OT Approx. w/ Adaptive Temp. | $O(NM(d+L))$ | Low-Medium | Medium-High | Soft one-to-one | EMD-like properties; Differentiable; Adaptive temp. (no global $\epsilon$ tuning) | Higher cost than CD; Approx. quality (vs exact EMD) |

The dominant computational costs of APML arise from the pairwise distance computation, with complexity $O(NMd)$, and the iterative Sinkhorn normalization, which requires $L$ matrix scaling steps. This results in an overall complexity of $O(NM(d+L))$. For point clouds of low dimensionality (e.g., $d = 3$) and fixed $L$ (e.g., $L = 20$), the total cost remains within a practical regime. In contrast to exact EMD solvers, APML avoids combinatorial optimization by relying on differentiable scaling operations, making it compatible with standard backpropagation pipelines.

The adaptive temperature mechanism further differentiates APML from other methods that require manual tuning of regularization parameters, such as $\epsilon$ in entropy-regularized Sinkhorn distances or $\alpha$ in HyperCD. This adaptivity provides per-instance control over the sharpness of the assignment, enabling the loss to adjust locally to the underlying geometry of the point sets. This design aims to preserve stable gradients and promote better structural alignment in both dense and sparse regions.

By integrating a transport-based formulation with adaptive regularization, APML represents a principled alternative to existing point-based losses, combining efficiency, flexibility, and improved theoretical properties for learning-based geometric matching.

# B    Technical Appendix: Description of datasets and metrics

This appendix describes in more detail the datasets used and the evaluation metrics chosen.

## B.1    Evaluation Databases

The evaluation is conducted using three datasets, which cover standard benchmarks for point cloud completion and a modality-specific task involving generation from WiFi-based measurements:

- **PCN Dataset [33]:** The PCN dataset is a commonly used benchmark derived from ShapeNet-Core [6]. It includes pairs of partial and complete 3D point clouds for a set of object categories such as airplane, cabinet, car, chair, lamp, sofa, table, and watercraft. The partial inputs typically contain 2048 points, while the ground truth shapes consist of 16384 points. These data pairs are used to assess the ability of a model to recover full geometry from incomplete observations. For the PCN evaluation we follow their standard official splits and report results on the test sets.

- **ShapeNet (SN34/SN55) [6]:** The ShapeNet34 and ShapeNet55 subsets are drawn from the ShapeNetCore dataset [6], including 34 and 55 object categories, respectively. These subsets are used to test model generalization in point cloud completion tasks under more challenging conditions. for ShapeNet-34 and ShapeNet-55, we follow the *hard* evaluation protocol, which measures generalization to unseen categories and shapes. ShapeNet-55 includes all 55 categories for both training and evaluation. ShapeNet-34 is trained and tested on a 34-category subset, while *ShapeNet Unseen-21* evaluates generalization by training on the same 34 categories but testing on the remaining 21. For the PCN and MM-Fi datasets, we follow their standard official splits and report results on the test sets.

- **MM-Fi Dataset [30]:** To evaluate APML on a point cloud generation task from a different modality, we use the MM-Fi dataset [30]. This dataset provides WiFi Channel State Information (CSI) measurements collected from commercial WiFi devices, along with corresponding ground truth 3D point clouds captured with a LiDAR device representing human subjects performing various activities in indoor settings. This dataset is particularly relevant for assessing the robustness of loss functions in scenarios involving noisy, real-world sensor data and the generation of complex, dynamic human shapes. For MM-Fi datasets, we follow their standard official splits and report results on the test sets.

- **PCOU3D Synthetic Dataset [5]:** For the additional experiments, we constructed a synthetic dataset, as shown in Figure 3, as a controlled environment to evaluate the stability of APML under different input conditions. The dataset comprises four analytic primitives: a cube, a sphere, a pyramid, and a Gaussian blob. Each primitive is represented as a point cloud with 1024 points sampled uniformly from the interior of the shape within the interval $[-0.5, 0.5]^3$. The data are partitioned into 1,600 shapes for training, 400 for validation, and 400 for testing, with all ground-truth point clouds containing exactly 1024 points. During training, the encoder receives a modified version of the ground truth, while the decoder is supervised against the complete 1024-point cloud. Five regimes are considered: A) all sparse (256 points), B) all dense (1024 points), C) mixed batches with 50% sparse and 50% dense, D) random assignment with $p(\text{sparse}) = 0.5$, and E) completion with a contiguous patch removed, leaving 512–1023 points. Regimes A–D evaluate reconstruction robustness to density variations, while regime E examines completion with missing surface regions. This setup enables systematic evaluation of stability and density sensitivity across controlled conditions.

## B.2  Evaluation Metrics

To quantitatively assess and compare the performance of models trained with different loss functions for point cloud completion and generation tasks, we adopt the following standard metrics. Let $\hat{X} = \{\hat{x}_1, \ldots, \hat{x}_N\}$ be the predicted point set and $X = \{x_1, \ldots, x_M\}$ be the ground truth point set.

**Chamfer Distance (CD).** The Chamfer Distance measures the average nearest-neighbor distance between two point sets [10]. We report two common variants:

- **CD-L1 (Mean $L_2$ distances):** This is the sum of average Euclidean distances between each point in one set and its closest point in the other set. Lower values are better.

$$\mathcal{L}_{\text{CD-L1}}(\hat{X}, X) = \frac{1}{N} \sum_{\hat{x} \in \hat{X}} \min_{x \in X} \|\hat{x} - x\|_2 + \frac{1}{M} \sum_{x \in X} \min_{\hat{x} \in \hat{X}} \|x - \hat{x}\|_2. \tag{10}$$

- **CD-L2 (Mean Squared $L_2$ distances):** This is the sum of average squared Euclidean distances, and is the most common CD formulation. Lower values are better.

$$\mathcal{L}_{\text{CD-L2}}(\hat{X}, X) = \frac{1}{N} \sum_{\hat{x} \in \hat{X}} \min_{x \in X} \|\hat{x} - x\|_2^2 + \frac{1}{M} \sum_{x \in X} \min_{\hat{x} \in \hat{X}} \|x - \hat{x}\|_2^2. \tag{11}$$

**Earth Mover's Distance (EMD).** EMD measures the minimum cost to transform one point set into another, reflecting overall structural similarity [22]. For point sets $\hat{X}$ and $X$ of equal cardinality ($N = M$), it is defined as the solution to an optimal assignment problem:

$$\mathcal{L}_{\text{EMD}}(\hat{X}, X) = \min_{\phi: \hat{X} \to X} \sum_{\hat{x}_i \in \hat{X}} \|\hat{x}_i - \phi(\hat{x}_i)\|_2, \tag{12}$$

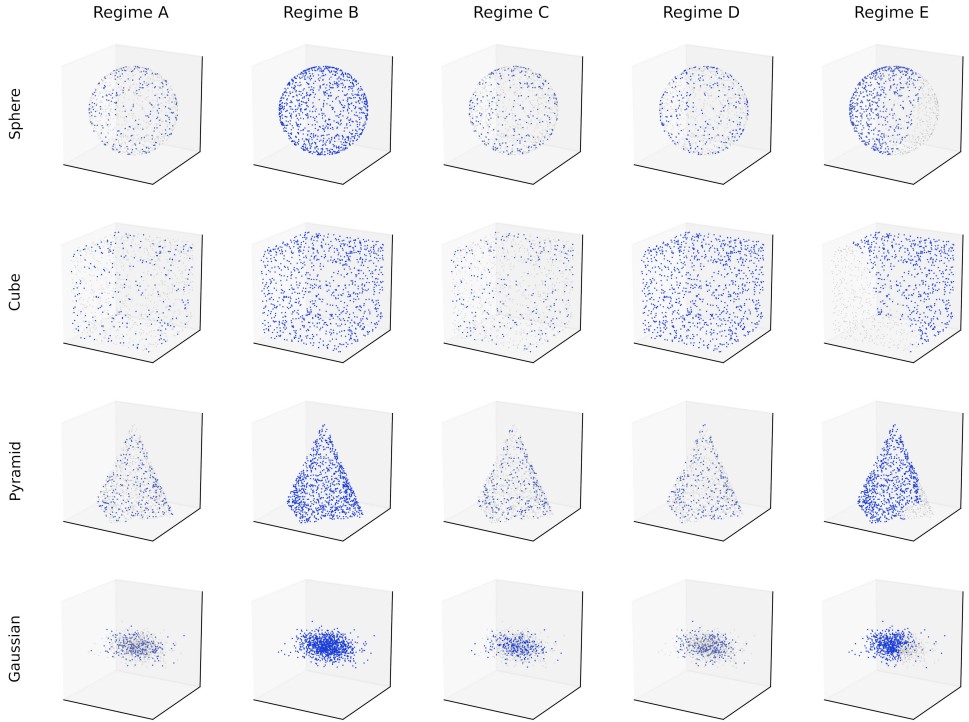

Figure 3: Qualitative visualization on PCOU3D. Rows correspond to four analytic shapes: sphere, cube, pyramid, and Gaussian cluster. Columns correspond to the five input regimes used in our stability study: A sparse input with 256 points, B dense input with 1024 points, C batch level mix of sparse and dense, D random per sample mix with probability 0.5 for sparse, and E completion where a contiguous patch is removed leaving between 512 and 1023 points. In each panel the full ground truth point cloud is shown in light gray and the regime specific input is overlaid in blue.

where $\phi$ is a bijection. Due to its computational cost and cardinality constraint (implementations often require sampling or padding if $N \neq M$), we report EMD multiplied by 100 (EMD$\times$100). Lower values are better.

**F1-Score (@$\tau$).** The F1-score assesses reconstruction accuracy by balancing precision and recall, commonly used in point cloud completion [33]. Given a distance threshold $\tau$, precision $P(\tau)$ and recall $R(\tau)$ are defined as:

$$P(\tau) = \frac{1}{N} \sum_{\hat{x} \in \hat{X}} \mathbb{I}\left(\min_{x \in X} \|\hat{x} - x\|_2 < \tau\right), \tag{13}$$

$$R(\tau) = \frac{1}{M} \sum_{x \in X} \mathbb{I}\left(\min_{\hat{x} \in \hat{X}} \|x - \hat{x}\|_2 < \tau\right), \tag{14}$$

where $\mathbb{I}(\cdot)$ is the indicator function. The F1-score is their harmonic mean:

$$F1(\tau) = 2 \cdot \frac{P(\tau) \cdot R(\tau)}{P(\tau) + R(\tau) + \varepsilon_{\text{F1}}}, \tag{15}$$

where $\varepsilon_{\text{F1}}$ is a small constant (e.g., $10^{-8}$) to prevent division by zero if both $P(\tau)$ and $R(\tau)$ are zero. In our experiments, we use a threshold value of $\tau = 0.01$. Higher F1-scores are better.

We note that while CD-L1 is widely used in evaluation, its suitability for accurately reflecting perceptual quality in generation tasks can be limited. As will be discussed with our qualitative results, numerically higher CD values do not always correspond to visually worse point clouds, particularly when distributed soft matching is involved, and vice-versa.

# C   Technical Appendix: Disaggregated results and statistical significance

This appendix shows disaggregated results per category and complementary metrics for some of the experiments in Section 4. We evaluated APML on the point cloud completion task using the PCN dataset, comparing its performance when integrated into three different backbone architectures, FoldingNet [31], PCN [33], and PoinTr [32], against models trained with HyperCD [17] as a strong baseline. We present the results in Table 4.

Table 4: Performance across PCN Dataset using Metric EMD*100, disaggregated by categories. Best results per backbone highlighted.

| | PCN Object Categories | | | | | | | |
| --- | --- | --- | --- | --- | --- | --- | --- | --- |
| | Airplane | Cabinet | Car | Chair | Lamp | Sofa | Table | Watercraft |
| FoldingNet + CD | 15.20 | 35.98 | 23.26 | 36.30 | 38.32 | 30.16 | 28.43 | 22.01 |
| FoldingNet + HCD | 14.19 | 27.03 | 19.30 | 30.26 | 30.86 | 23.73 | 19.52 | 18.55 |
| FoldingNet + InfoCD | 13.76 | 4.15 | 18.41 | 23.27 | 28.54 | 22.91 | 21.42 | 17.42 |
| FoldingNet + APML | **3.36** | **5.62** | **3.84** | **6.40** | **7.89** | **5.41** | **5.66** | **4.52** |
| PCN + CD | 34.70 | 16.82 | 10.56 | 14.66 | 19.84 | 15.09 | 10.79 | 10.46 |
| PCN + HCD | 3.86 | 6.61 | 3.91 | 6.81 | 13.21 | 6.02 | 6.52 | 5.31 |
| PCN + InfoCD | 3.54 | 5.83 | 3.58 | 5.40 | 10.01 | 5.66 | 5.71 | 4.54 |
| PCN + APML | **3.14** | **4.41** | **3.33** | **4.83** | **8.71** | **4.59** | **4.76** | **4.00** |
| PoinTr + CD | 5.62 | 9.16 | 6.47 | 8.29 | 20.28 | 9.58 | 9.28 | 6.96 |
| PoinTr + HCD | 4.78 | 9.05 | 6.19 | 7.99 | 18.40 | 9.36 | 8.47 | 6.06 |
| PoinTr + InfoCD | 8.31 | 9.88 | 8.76 | 9.83 | 11.29 | 11.22 | 11.07 | 7.39 |
| PoinTr + APML | **3.22** | **6.14** | **5.30** | **5.55** | **7.49** | **6.03** | **6.25** | **5.02** |

## C.1   Stability Analysis on the PCOU3D Dataset

To assess the stability of APML under controlled conditions, we introduce the PCOU3D synthetic dataset described in Appendix B.1. The dataset allows reproducible experiments across five distinct regimes: A (sparse, 256 points), B (dense, 1024 points), C (batch-level mix 50% A + 50% B), D (random per-sample mix $p = 0.5$), and E (completion with contiguous patch removal, 512–1023 points).

We trained FoldingNet for 30 epochs using identical hyperparameters for all losses (Adam optimizer, learning rate $1 \times 10^{-4}$, batch size 32). Each model was supervised with the complete 1024-point cloud. Results are reported on the test set, and lower values indicate better performance.

Table 5: Comparison of performance using CD and EMD metrics on the PCOU3D synthetic dataset. Lower values are better.

| Loss | CD (↓) | | | | | EMD (↓) | | | | |
| --- | --- | --- | --- | --- | --- | --- | --- | --- | --- | --- |
| | A | B | C | D | E | A | B | C | D | E |
| APML | **0.0474** | 0.0497 | **0.0502** | **0.0471** | **0.0481** | **0.0433** | **0.0459** | **0.0485** | **0.0416** | **0.0423** |
| CD | 0.0529 | **0.0413** | 0.0429 | 0.0495 | 0.0492 | 0.0706 | 0.0585 | 0.0626 | 0.0600 | 0.0583 |
| InfoCD | 0.0496 | 0.0557 | 0.0495 | 0.0509 | 0.0495 | 0.0578 | 0.0805 | 0.0597 | 0.0641 | 0.0639 |

**Key observations.**

Table 5 summarizes the stability analysis across all regimes. APML achieves consistently lower losses and minimal variation between conditions. For example, the difference between the best and worst regime in EMD remains below 5%, indicating a high robustness compared to CD and InfoCD. Training with sparse-only inputs (regime A) increases the EMD for CD by approximately 17% due to many-to-one assignments, while APML shows a variation less than 6%. This confirms that APML maintains stable matching behavior even under strong input sparsity.

This stability originates from the adaptive temperature $T$, computed locally for each row of the cost matrix based on the distance gap $g$ (see Eq. 3). When candidate targets are close (e.g., distances 0.010 and 0.012, $g = 0.002$), $T$ becomes large ($T \approx 80$), producing a soft assignment $P \approx (0.83, 0.15, \dots)$. When the best match is clear (e.g., distances 0.010 and 0.40, $g = 0.39$), $T$ becomes small ($T \approx 0.4$), yielding a sharp $P \approx (0.99, 0.006, \dots)$. As $T$ is recomputed for each mini-batch, ambiguous matches remain soft while confident matches stay sharp, ensuring smooth gradients regardless of density or sampling.

We also tested the effect of disabling the uniform fallback, which replaces the temperature-scaled softmax with a uniform distribution when several minima coincide ($g < \epsilon_g$). Removing this safeguard substantially worsened performance, raising the EMD on dense inputs from approximately 0.055 to 0.10, and confirming its importance for numerical stability when ambiguous matches occur. These results demonstrate that the analytically derived temperature adapts automatically to local structure, avoiding the need for a manually tuned hyperparameter and maintaining stability across all sampling regimes. Qualitative reconstructions for regime A are illustrated in Fig. 4, showing that APML retains global shape integrity across all primitives while CD and InfoCD shows local collapses or flattened regions.

### C.2 Ablation Studies on $p_{\min}$ and APML Components

Table 6 reports an extended sweep of $p_{\min}$ spanning four orders of magnitude ($10^{-4}$ to 0.999) on the PCOU3D dataset, together with the corresponding results on MM-Fi for the tested values. Across all regimes (sparse input (A), dense input (B), and completion (E)), CD and EMD remain nearly invariant within the practical range $p_{\min} \in [0.01, 0.8]$, with variations below 5%. The same stability pattern appears on MM-Fi, confirming that APML is largely insensitive to this hyperparameter.

Table 6: Ablation of $p_{\min}$ on PCOU3D and MM-Fi. Lower is better. Bold values denote the best performance in each regime.

| $p_{\min}$ | A (Sparse) | | B (Dense) | | E (Completion) | | MM-Fi | |
|---|---|---|---|---|---|---|---|---|
| | CD | EMD | CD | EMD | CD | EMD | CD | EMD |
| 0.0001 | 0.05335 | 0.04722 | 0.05973 | 0.06322 | 0.05302 | 0.04809 | – | – |
| 0.01 | 0.05283 | 0.05248 | 0.05703 | 0.05559 | 0.05324 | 0.05648 | 0.1431 | 0.1686 |
| 0.1 | 0.05068 | 0.04954 | 0.05372 | 0.05907 | 0.05505 | 0.05273 | 0.1398 | 0.1661 |
| 0.25 | 0.05466 | 0.06696 | 0.05653 | 0.05586 | 0.05346 | 0.04937 | – | – |
| 0.5 | 0.05586 | **0.04945** | **0.05033** | **0.04300** | 0.05417 | 0.05093 | **0.1371** | 0.1671 |
| 0.75 | 0.05655 | 0.05121 | 0.06147 | 0.05975 | 0.05480 | 0.05094 | – | – |
| 0.8 | 0.05331 | 0.05067 | 0.05443 | 0.05191 | 0.05840 | 0.05514 | 0.1392 | **0.1637** |
| 0.9 | **0.05067** | 0.05784 | 0.06168 | 0.06910 | **0.05286** | **0.04892** | – | – |
| 0.999 | 0.05262 | 0.05040 | 0.05618 | 0.05372 | 0.05326 | 0.05200 | – | – |

This behaviour can be explained analytically. In APML, $p_{\min}$ acts only as a lower bound on the probability assigned to the closest candidate in the adaptive softmax. The actual sharpness of the assignment is determined by the adaptive temperature $T$, calculated from the local distance gap $g$ as shown in Equation 3. The mapping $T(p_{\min})$ is monotonic and smooth for all $0 < p_{\min} < 1$. For small $p_{\min}$, $T$ grows approximately linearly, creating softer assignments. For larger $p_{\min}$, $T$ saturates, producing sharper distributions. Because $T$ is recomputed independently for every row and mini-batch, this self-adaptive mechanism ensures that local geometric relations dominate the matching behaviour rather than the global hyperparameter setting. Consequently, changes in $p_{\min}$ across several decades do not substantially alter the resulting transport plan or its gradients. The observed robustness reflects that the coupling between $T$ and $g$ maintains numerical stability and consistent matching entropy without the need for manual tuning.

**Ablation of APML components.** We further isolate the contributions of each component of the APML formulation: (1) *Full*: complete APML with adaptive temperature, symmetrization, and Sinkhorn normalization; (2) *No Sinkhorn*: adaptive temperature and symmetrization but without the iterative normalization; (3) *Sinkhorn-only*: fixed temperature ($T = 0.1$) without adaptivity; (4) *No*

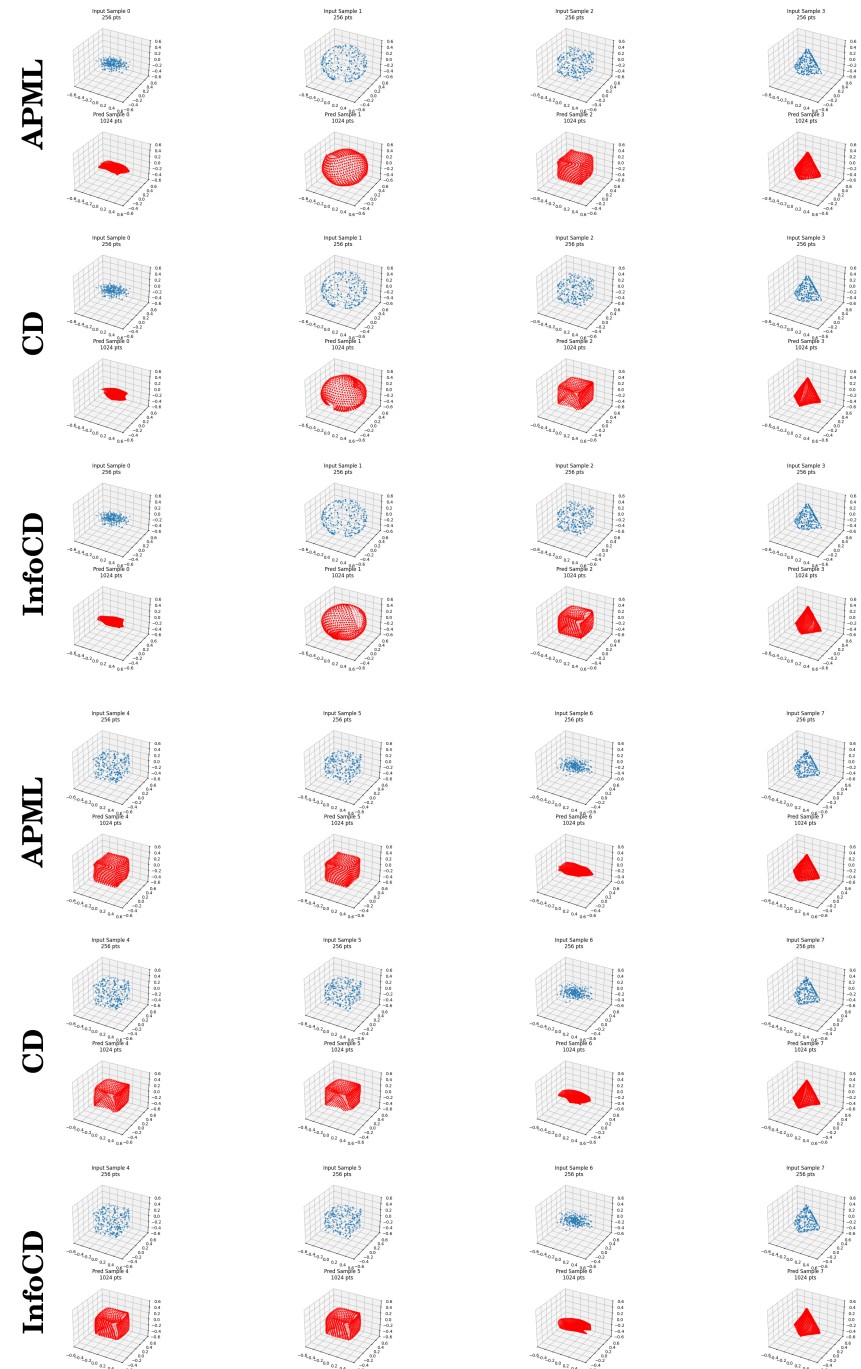

Figure 4: Qualitative comparison on the PCOU3D synthetic dataset under sparse-input regime (A). Each column shows an input shape (256 points, blue) and the corresponding predicted dense reconstruction (1024 points, red) for different analytic primitives (sphere, cube, pyramid, Gaussian cluster). The rows correspond to models trained with APML, standard CD, and InfoCD losses. APML produces geometrically faithful reconstructions that better preserve global structure and surface continuity, even when the input is sparse and incomplete. Visual results align with the quantitative findings in Table 5, confirming that the adaptive temperature stabilizes optimization across sampling regimes.

*symmetrization*: unidirectional matching without averaging the forward and backward assignments. The results are presented in Table 7.

Table 7: Ablation of APML components on the PCOU3D dataset under regimes A (sparse), B (dense), and E (completion). Lower is better.

| Configuration | A (Sparse) | | B (Dense) | | E (Completion) | |
|---|---|---|---|---|---|---|
| | CD | EMD | CD | EMD | CD | EMD |
| Full | **0.0531** | **0.0519** | **0.0544** | **0.0519** | **0.0584** | **0.0551** |
| No Sinkhorn | 0.0588 | 0.0639 | 0.0659 | 0.0814 | 0.0557 | 0.0851 |
| Sinkhorn-only ($T$=0.1) | 0.1367 | 0.1029 | 0.1439 | 0.1081 | 0.1508 | 0.1161 |
| No symmetrization | 0.0571 | 0.0538 | 0.0582 | 0.0555 | 0.4788 | 0.4122 |

The results confirm that each stage of APML is functionally necessary. Removing the Sinkhorn normalization disrupts the marginal constraints of the transport plan, producing biased probability mass accumulation and increased EMD, especially under dense and completion regimes. Using a fixed temperature ($T = 0.1$) without adaptivity prevents the softmax from scaling with the local distance gap, leading to unstable optimization and loss of geometric consistency. The absence of symmetrization introduces directional bias, severely degrading the completion regime due to one-sided correspondences. The full configuration preserves both geometric fidelity and numerical stability, demonstrating that the adaptive temperature and Sinkhorn normalization jointly define the probabilistic matching process. These findings support the theoretical interpretation that $T$ shapes the initial probability landscape, while Sinkhorn refinement ensures coherent transport consistency between predicted and ground truth sets.

### C.3 Statistical significance

We perform statistical significance tests between different loss functions. Figure 5 reports two-sided Wilcoxon signed-rank tests across the eight PCN categories. APML differs significantly from every Chamfer-style loss ($p = 0.008$ in all comparisons), corroborating the quantitative gains in Table 1. In contrast, the gap between InfoCD and HyperCD is not significant ($p = 0.078$), matching their very similar mean F1 scores. These results support the claim that APML delivers a systematic, category-wise improvement rather than isolated wins in a few classes.

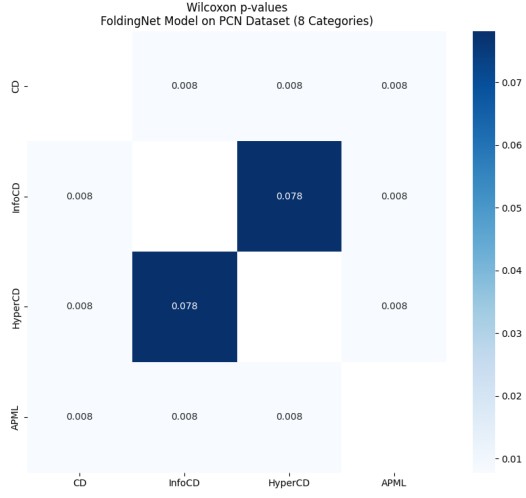

Figure 5: Pair-wise Wilcoxon signed-rank $p$-values (8 object categories) comparing class-wise F1 scores for FoldingNet on the PCN dataset. Darker cells denote higher $p$; values below the 0.05 diagonal line indicate statistically significant differences.

## C.4   Results with additional metrics

We focus our main discussion on the Earth Mover's Distance (EMD) results, as EMD is often considered a more reliable indicator of perceptual and structural similarity. However, we provide comprehensive results on various metrics including F1-score (Table 8), CD-L1 (Table 9), and CD-L2 (Table 10).

Table 8: Performance across PCN Dataset Metric F1, disaggregated by categories

|  | airplane | cabinet | car | chair | lamp | sofa | table | watercraft |
|---|---|---|---|---|---|---|---|---|
| FoldingNet + HCD | 0.787 | 0.436 | 0.571 | 0.411 | 0.439 | 0.389 | 0.571 | 0.573 |
| FoldingNet + APML | 0.773 | 0.535 | 0.598 | 0.473 | 0.477 | 0.464 | 0.631 | 0.589 |
| PCN + HCD | 0.863 | 0.581 | 0.683 | 0.542 | 0.570 | 0.520 | 0.706 | 0.679 |
| PCN + APML | 0.842 | 0.566 | 0.662 | 0.534 | 0.558 | 0.496 | 0.681 | 0.653 |
| PoinTr + HCD | 0.929 | 0.677 | 0.733 | 0.737 | 0.825 | 0.652 | 0.823 | 0.821 |
| PoinTr + APML | 0.864 | 0.548 | 0.638 | 0.627 | 0.700 | 0.545 | 0.727 | 0.718 |

Table 9: Performance across PCN Dataset Metric CD-L1, disaggregated by categories

|  | airplane | cabinet | car | chair | lamp | sofa | table | watercraft |
|---|---|---|---|---|---|---|---|---|
| FoldingNet + HCD | 7.601 | 12.657 | 10.484 | 13.492 | 12.977 | 13.320 | 11.338 | 10.988 |
| FoldingNet + APML | 8.223 | 13.391 | 11.070 | 15.486 | 15.913 | 15.066 | 12.414 | 12.007 |
| PCN + HCD | 6.076 | 11.897 | 9.470 | 12.645 | 12.624 | 12.938 | 9.932 | 9.984 |
| PCN + APML | 6.631 | 13.101 | 10.091 | 13.676 | 13.774 | 14.550 | 11.082 | 10.674 |
| PoinTr + HCD | 4.589 | 9.693 | 8.361 | 8.377 | 6.822 | 9.715 | 7.045 | 6.678 |
| PoinTr + APML | 5.957 | 11.813 | 9.865 | 10.595 | 9.371 | 12.450 | 9.103 | 8.622 |

Table 10: Performance across PCN Dataset Metric CD-L2, disaggregated by categories

|  | airplane | cabinet | car | chair | lamp | sofa | table | watercraft |
|---|---|---|---|---|---|---|---|---|
| FoldingNet + HCD | 0.249 | 0.530 | 0.338 | 0.691 | 0.692 | 0.678 | 0.595 | 0.456 |
| FoldingNet + APML | 0.348 | 0.695 | 0.431 | 1.037 | 1.134 | 0.987 | 0.825 | 0.612 |
| PCN + HCD | 0.172 | 0.522 | 0.292 | 0.619 | 0.725 | 0.678 | 0.477 | 0.420 |
| PCN + APML | 0.238 | 0.750 | 0.352 | 0.774 | 0.936 | 0.897 | 0.622 | 0.493 |
| PoinTr + HCD | 0.095 | 0.358 | 0.231 | 0.267 | 0.209 | 0.346 | 0.208 | 0.165 |
| PoinTr + APML | 0.161 | 0.514 | 0.333 | 0.477 | 0.473 | 0.796 | 0.399 | 0.299 |

As shown in Table 4, models trained with APML consistently achieve lower EMD scores compared to those trained with HyperCD across the three evaluated architectures. This trend is observed in most category of objects. For example, when using FoldingNet, the EMD score for the category 'airplane' is reduced from 14.19 to 3.36, and for 'lamp' from 30.86 to 7.89. Similar reductions are observed with the PCN and PoinTr architectures, indicating that the predictions obtained with APML are more geometrically aligned with the ground truth under the EMD metric.

When using CD-L1, CD-L2, and F1-score for evaluation (see Appendix A), the differences are less significant. In some cases, models trained with HyperCD yield lower CD-based errors. This is expected, given that the loss function directly optimizes a Chamfer-based objective and therefore induces a bias in favor of CD-like metrics. Although APML draws from the same theoretical framework as EMD, it does not directly minimize the EMD score during training. However, its performance according to this metric indicates an improvement in the structural consistency between the predicted and reference point clouds. These results are consistent with the qualitative observations presented in Figure 1, where APML leads to more coherent spatial reconstructions, smoother point distributions, and more detailed completions.

# D   Technical Appendix: Additional experiments and visualizations

This appendix provides additional visualizations of the experiments and metrics.

## D.1   Qualitative Comparison: Visual vs. Metric Discrepancy

Figure 6 illustrates a representative case study comparing point cloud reconstructions from four loss functions: CD, InfoCD, HyperCD, and our proposed APML. All outputs are generated using the same FoldingNet backbone trained on the PCN dataset.

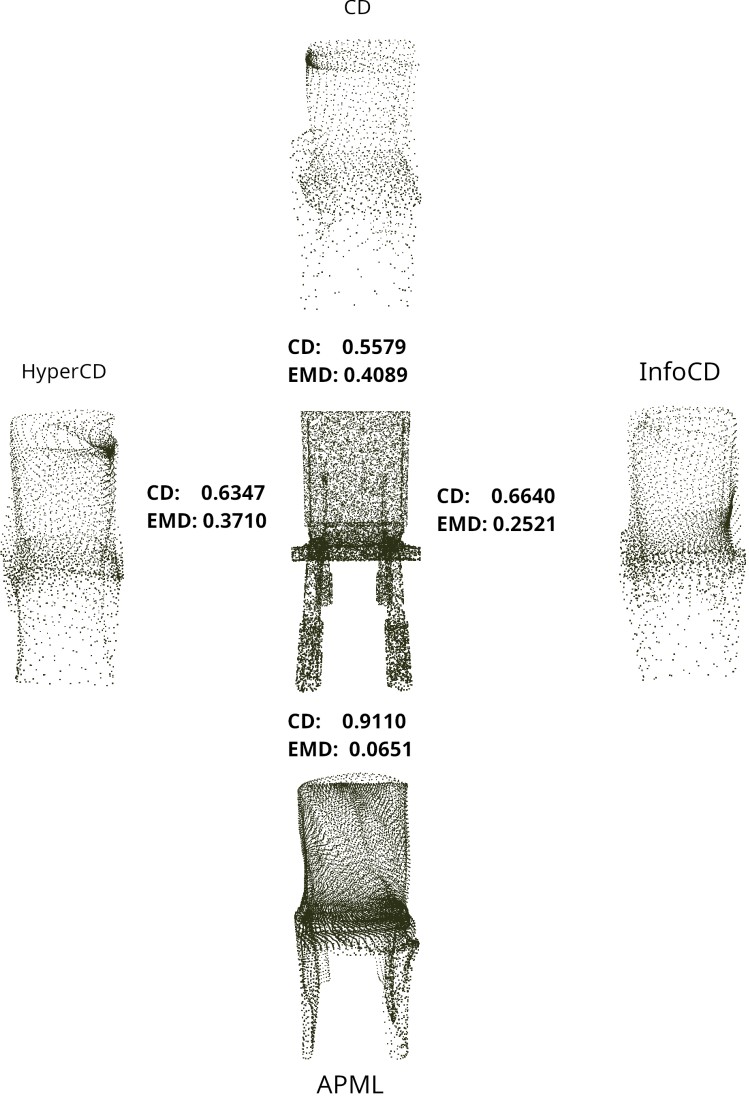

Figure 6: Qualitative comparison of chair reconstructions across different loss functions. All models are trained using FoldingNet on the PCN dataset. Despite lower Chamfer Distance (CD) values, CD- and InfoCD-based reconstructions suffer from visible structural artifacts. APML yields superior perceptual quality and shape integrity, aligning better with the ground truth (center).

Although CD and InfoCD achieve lower Chamfer Distance scores, the reconstructions are perceptually inferior, showing missing legs, clumping, or collapsed surfaces. HyperCD slightly improves both CD and Earth Mover's Distance (EMD), but artifacts remain visible. In contrast, APML produces a clean

and geometrically plausible reconstruction, despite yielding a higher CD. This example underscores a known limitation of CD-based metrics: they disproportionately emphasize dense regions and fail to penalize structural mismatches. EMD, although more expensive to compute, better reflects perceptual alignment. APML minimizes this gap by promoting soft, one-to-one correspondences, improving EMD and qualitative fidelity simultaneously. These results support our broader claim: CD alone is insufficient to evaluate reconstruction quality, and APML offers a more robust supervision signal for geometry-aware learning.

To further assess generalization across architectures and datasets, we extended the qualitative analysis to the Transformer-based PoinTr model trained on the ShapeNet-55 dataset. Figure 7 presents an example of car reconstruction results using CD, InfoCD, HyperCD, and APML losses.

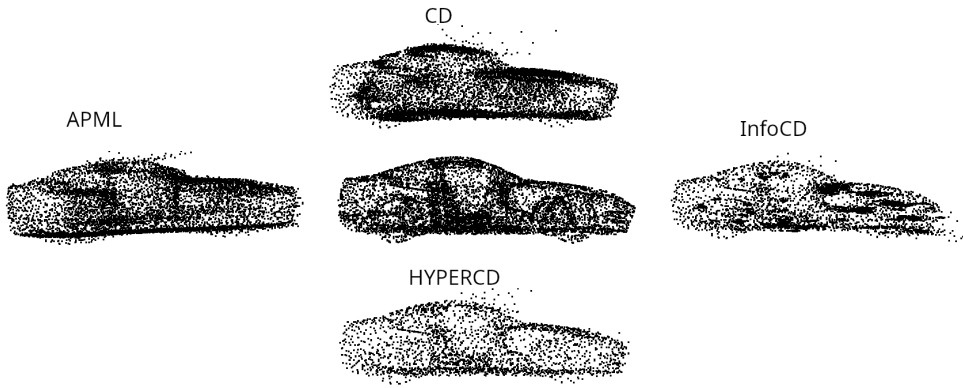

Figure 7: Qualitative comparison of car reconstructions using the PoinTr backbone on the ShapeNet-55 dataset. CD and InfoCD generate oversmoothed or partially collapsed geometries, particularly around the roof and rear. HyperCD improves global outline but introduces uneven surface density. APML produces the most coherent and well-defined reconstruction, preserving both large-scale structure and fine details such as the hood and windshield curvature.

The visual comparison in Figure 7 highlights that APML maintains high geometric consistency even in more complex models where attention mechanisms amplify local feature dependencies. The probabilistic formulation provides stable gradients during training and avoids degenerate clustering in dense regions, a common issue in CD-based methods. These qualitative results confirm that APML not only improves EMD metrics but also enhances the perceptual realism of reconstructed shapes, validating its robustness across architectures and datasets.

## D.2 Convergence Analysis

To better understand the optimization dynamics of APML compared to other loss functions, we visualize the training convergence behavior in terms of F1 score over epochs. Figure 8 shows the evolution of F1 score on the validation set during training for to representative configurations: the PCN backbone trained on the PCN dataset, and the FoldingNet backbone trained on SN34 and SN55.

As shown in the figure, APML consistently achieves faster convergence compared to other methods. Within the first 20 epochs, it significantly outpaces CD and its variants, and maintains a stable lead in final performance. The curve for APML shows both higher peak F1 and reduced variance during late-stage training, suggesting that the smooth gradients and one-to-one soft matching induced by our loss improve both optimization speed and stability.

HyperCD and InfoCD, while outperforming CD, still lag behind APML throughout training. These results support the claim that APML accelerates convergence by reducing ambiguity in point assignments and encouraging better spatial regularity in the predicted clouds.

## D.3 Empirical Sparsity of the Transport Matrix

We conduct empirical studies on the sparsity of the transport matrix, as constructed before Sinkhorn. Across all training batches we find that fewer than $8\%$ of the entries in $P$ exceed $10^{-3}$, implying

**effective sparsity well above 90 %**. The pattern in Figure 9 (computed for the smaller CSI2PC / MM-Fi model) is typical for all our transport matrices: most probability mass concentrates on a single column (value 1), a small shoulder appears near the adaptive gap ($\approx 0.5$ here), and the remainder of the row is near-zero.

Each heat-map shows a $1\,\mathrm{k} \times 1\,\mathrm{k}$ crop from a $1.2\,\mathrm{k} \times 1.2\,\mathrm{k}$ frame block. The bright vertical stripe corresponds to the single high-probability match selected by the adaptive softmax; faint horizontal traces stem from the bidirectional averaging step (Sec. 3.2). All remaining cells are exactly zero after thresholding at $10^{-4}$. Again, the patters are also typical for bigger matrices in larger models (up to $16\,\mathrm{k} \times 16\,\mathrm{k}$ frame blocks).

**Implications for memory.** Storing $P$ densely dominates GPU memory for large point counts. The empirical sparsity suggests two straightforward mitigations:

1. **Thresholded sparse format.** Retaining only values $\geq 10^{-3}$ yields a $\sim$5–6$\times$ reduction in practice, bringing the largest $16\,\mathrm{k}$ matrix below the $40\,\mathrm{GB}$ limit of a single A100.

2. **Mixed precision.** Encoding the surviving non-zero blocks in FP16 would shave a further $\sim$30 % off the footprint.

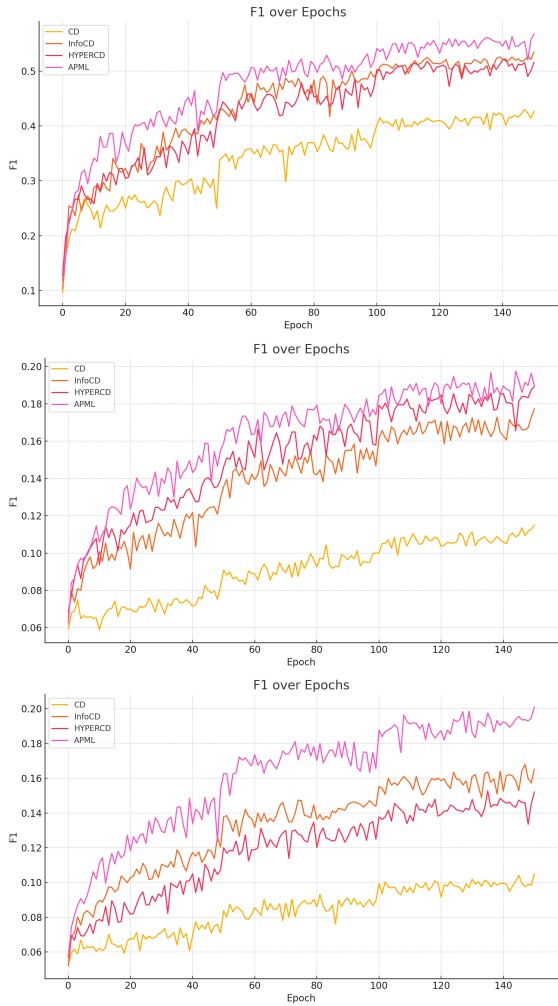

Figure 8: F1 score on the validation set (**Top:** PCN. **Middle:** SN34 **Bottom:** SN55) over 150 epochs for four loss functions: Chamfer Distance (CD), InfoCD, HyperCD, and our proposed APML. The backbone is FoldingNet trained on the PCN dataset.

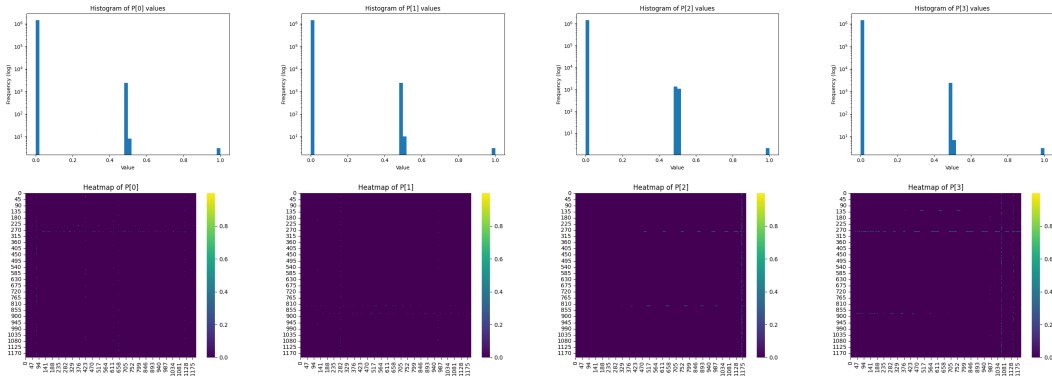

Figure 9: Examples of the sparsity of the transport matrix $P$ ($1.2\text{k} \times 1.2\text{k}$), as computed before Sinkhorn for the CSI2PC model. **Top four:** log–frequency histograms of four randomly selected frames (all bins left of 0.05 are clamped into the first bar). **Bottom four:** $1\text{k} \times 1\text{k}$ crops of the same frames visualised as heat-maps (log colour-scale). In every row, more than 90% of entries are smaller than $10^{-3}$, producing a dominant spike at zero and confirming that $P$ is effectively sparse.

Both the histogram and heat-map example views confirm that *effective* memory usage is dominated by a tiny subset of matrix entries. This justifies storing $P$ in a sparse COO/CSR format or computing with block-sparse kernels, reducing peak GPU RAM by an order of magnitude without altering the optimisation dynamics.

**Sparse implementation of APML**. Based on these findings, we implemented a sparsity-based optimization in which, dynamically, over $99\%$ of the less important cost values are culled (set to zero), yielding practically the same empirical results.

Our experiments show that the Sparse APML variant achieves close to linear memory scaling and up to a $99\%$ reduction in memory usage compared to the dense baseline. To show that quality changes are minimal, we trained CSI2PC on MM-FI with the `manual_split` configuration from the original source for 100 epochs and compare the results in Table 11.

Table 11: Loss comparison on MM-FI (Comparing Sparse implementation).

| Dataset | Model | Loss | CD ($\downarrow$) | EMD ($\downarrow$) |
|---------|-------|------|------|-------|
| MM-FI | CSI2PC | Sparse APML | 0.148 | 15.90 |
| MM-FI | CSI2PC | Dense APML | 0.139 | 16.37 |

As expected, the results are similar, and the images generated with both are almost the same without a major change.

**Peak memory usage.** Table 12 summarizes the peak memory footprint for Sparse APML on a synthetic dataset and the per-sample reduction.

Table 12: Peak memory usage per sample vs. number of points.

| #Points | Dense APML | Sparse APML | Reduction |
|---------|------------|-------------|-----------|
| 1024 | 50 MB | 0.39 MB | $\sim 99.23\% \downarrow$ |
| 4096 | 411 MB | 1.31 MB | $\sim 99.68\% \downarrow$ |
| 8192 | 1.6 GB | 2.62 MB | $\sim 99.84\% \downarrow$ |
| 32768 | 18.5 GB | 8.58 MB | $\sim 99.95\% \downarrow$ |
| 65536 | 68 GB | 17.1 MB | $\sim 99.97\% \downarrow$ |

For a particular case (e.g., PoinTR on ShapeNet34; see Appendix D3), peak memory consumption during training can be reduced by a factor of $99.9\times$ (from $320\,\text{GB}$ of VRAM to $< 0.5\,\text{GB}$). This demonstrates that APML is practical for large-scale applications.

