# OpenReview forum: "APML: Adaptive Probabilistic Matching Loss for Robust 3D Point Cloud Reconstruction"
_NeurIPS.cc/2025/Conference — NeurIPS 2025 poster_

### Official Review · Reviewer_ahb3 · 2025-07-01

**Clarity:** 3
**Significance:** 3
**Originality:** 3
**Rating:** 4
**Confidence:** 3

**Summary:**

The authors propose a loss function for 3D point cloud shape completion and generation. This loss function is based on the Sinkhorn Distance, where it uses a temperature-scaled and data-driven parameter T to replace the control parameter $\lambda$ for the entropy value in the Sinkhorn Distance.

This loss function achieves near-quadratic runtime, comparable to Chamfer-based losses, and approximates the matching quality of losses based on the Earth Mover's Distance.

The paper performs quantitative and qualitative evaluations on point cloud completion (ShapeNet and PCN) and generation (MM-Fi) benchmarks.

**Questions:**

1. Add an ablation experiment for parameter T that the authors propose.

2. Add experiments based on EMD loss to directly compare the performance between the loss the authors propose and EMD loss.

3. Add experiments that show results when choosing different $p_{min}$ values and explain why to choose them and how they influence the 3D point cloud tasks.

4. How to choose the value of the data-driven parameter T that the authors propose? There are no experiments about it.

5. The loss that the authors propose is based on the Sinkhorn loss and modifies its control parameter $\lambda$. This paper should add experiments showing what will happen if there are different $\lambda$ values when they are used in 3D point cloud tasks to verify the necessity of the change.

**Ethical Concerns:**

["NO or VERY MINOR ethics concerns only"]

**Final Justification:**

With the additioanl experiemnts on EMD and parameter analysis, I am overall satisfied on this work, remaining positive.

**Limitations:**

yes

**Quality:**

3

**Strengths And Weaknesses:**

Strengths:

1. This paper proposes an Adaptive Probabilistic Matching Loss, which mitigates common shortcomings of losses based on Chamfer Distance, such as point clumping and density bias.
2. The qualitative evaluation of APML is nearly identical to losses based on Earth Mover's Distance, while APML runtime is lower than EMD. APML combines the advantages of CD and EMD while avoiding their disadvantages in 3D point cloud tasks.

Weaknesses:

1. This paper lacks a comparison with EMD-based losses, which are among the most closely related losses to the proposed one, and serve as the primary motivation.
2. This paper lacks an ablation experiment on the parameter T, which the authors propose to replace the control parameter $\lambda$ of the entropy value in the Sinkhorn Distance.
3. This paper lacks a detailed explanation about how to choose the value of parameter T and how the parameter T influences 3D point cloud tasks.

---

> ### Author Rebuttal · Authors · 2025-07-30
>
> Thank you for your helpful and constructive review. We think that the following experiments and clarifications can greatly improve the paper.
>
> ### **1. Comparison with EMD-based losses:**
> We fully agree that EMD is an important baseline. However, its high computational cost prevents its use at training scale. As a metric, it significantly slows down evaluation even on small batches. We attempted several EMD approximations (e.g., [3], [4]) but observed instability, especially in high-density settings like MM-FI. Qualitatively, our results are on par with EMD-style reconstructions. We include these visual comparisons in the supplementary material. Preliminary attempts at using EMD for training on even moderate-sized toy datasets confirmed the infeasibility due to long runtimes and memory saturation. This practical limitation supports the use of APML as a scalable, effective proxy.
> We will include additional quantitative and visual comparisons in the camera-ready version.
>
> To give some insights about other metrics we conducted experiments comparing SWD and DCD against APML on both ShapeNet34 and MM-FI, showing higher EMD values and reduced stability.
>
> **Table G. Loss comparison on ShapeNet34 and MM-FI (FoldingNet and PoinTr).**
>
> | Dataset     | Model       | Loss  | CD ↓   | EMD ↓  | F1 Score ↑ |
> |-------------|-------------|-------|--------|--------|------------|
> | ShapeNet34  | PoinTr      | DCD   | –      | 9.48   | 0.47       |
> | **ShapeNet34**  | **PinTr**  | **APML** | –      | **8.14**   | **0.50**     |
> | ShapeNet34  | FoldingNet  | DCD   | –      | 11.52  | 0.19       |
> | ShapeNet34  | FoldingNet  | SWD   | –      | 11.98  | 0.11       |
> | **ShapeNet34**  | **FoldingNet**  | **APML** | –      | **9.49**   | **0.20**     |
> | MM-FI       | CSI2PC  | DCD   | **0.148**  | 25.68  | –          |
> | **MM-FI**       | **CSI2PC**  | **APML** | 0.152 | **14.11** | –          |
>
> While DCD showed a better CD metric than APML, like other CD-based variants, it struggled to produce dense and coherent representations of the body.
>
> We explored EMD approximations (point-set-based methods SWD [4]) on two dataset. Training FoldingNet on ShapeNet34 with SWD, showed that APML has better results for all metrics. However, for MM-FI, the training was unstable. The generated images failed to generalize well in terms of both body structure and coherence, (i.e the generated images did not resemble the ground truth, and were not comparable to those of APML). It has to be noted that SWD imposes an equal number of points between ground truths and predictions, which is incompatible with some models. While a full evaluation is pending, we emphasize that APML achieves near-EMD qualitative behavior at much lower cost.
>
> ### **2. Ablation on T / (data-driven) parameter and effect of p_min hyperparameter:**
>
> T (or p_min) controls the entropy regularization in the Sinkhorn optimization. Our analytic formulation ensures that it scales appropriately with point cloud resolution and distance magnitude. As such, it avoids the need for manual tuning and makes APML stable across tasks and datasets. We emphasize that the entropy parameter is not heuristically set, but derived from point-wise distance statistics. This principled approach ensures consistency and robustness across input conditions.
> This means that the proposed temperature parameter T is computed analytically using dataset-level statistics (hence its data-driven nature) and cannot be chosen. However, the hyperparameter p_min has effect on the values that T can take. To test its effect, we conducted a toy experiment where p_min is varied broadly. Results show consistent performance across this range. Specifically we tested p_min values on MM-FI and found negligible performance differences. These observations confirm the robustness of APML to its sole hyperparameter.
>
> **Table B. p_min sweep on MMFI.**
> | p_min Value | CD ↓   | EMD ↓  |
> |-------------|--------|--------|
> | 0.01       | 0.1431 | 0.1686 |
> | 0.1         | 0.1398 | 0.1661 |
> | 0.5         | 0.1371 | 0.1671 |
> | 0.8         | 0.1392 | 0.1637 |
>
> Additional experiments on a toy dataset (see reviewer dY8D), show that across a extense range of p_min values and different sparsity configurations, show variations smaller than 1% in both CD and EMD metrics.
>
> We also plotted the resulting temperature T as a function of p_min (for a fixed dataset size of 8192 points). The relationship is smooth and weakly non-linear for most of the relevant range (p_min ∈ [0.1, 0.9]), reinforcing the stability of the learned matching under different settings. These results empirically validate the robustness and explain why the choice of p_min is not sensitive. We will show this plot in the camera ready version of the supplementary material.
>
>
> ### **3. Clarification of Sinkhorn design and hyperparameter p_min:**
> Thank you for this question. To clarify, we do not use the entropic-regularized variant of Sinkhorn that involves a fixed λ (or ϵ) parameter in log-space. Instead, our method applies the standard Sinkhorn algorithm that alternates row and column normalization in the primal (cost) space, commonly referred to as Sinkhorn-Knopp normalization. In this setting, the softness of the resulting assignment matrix is controlled by scaling the input cost matrix before normalization.
>
> In APML, we modulate this scaling via a data-driven temperature parameter T (as explained before), computed analytically from local distance gaps. This approach eliminates the need for a global hyperparameter and allows the method to adapt to each point's levels of ambiguity: large cost gaps produce sharp assignments, while small gaps yield softer distributions.
>
> This confirms that the data-driven temperature mechanism is not a cosmetic change but a necessary design component for stable and generalizable matchings. As noted above, the single interpretable parameter in our method refers to p_min, which governs the adaptive scaling behavior.

---

> ### Comment · Reviewer_ahb3 · 2025-08-05
>
> Thank you for addressing my concerns. I am overall satisfied on this work, remaining positive.

---

> > ### Author Response · Authors · 2025-08-05
> >
> > We thank the reviewer for the interesting discussion and positive comments. We are happy to see that we were able to address their questions properly.

---

### Official Review · Reviewer_BHnV · 2025-07-01

**Clarity:** 3
**Significance:** 2
**Originality:** 2
**Rating:** 5
**Confidence:** 4

**Summary:**

This paper proposes using Adaptive Probabilistic Matching Loss (APML) as a loss function for point-cloud comparison. APML combines temperature-controlled point correspondences with the Sinkhorn algorithm to produce a matrix that is doubly stochastic. The temperature parameter introduced by the method is analytically determined, eliminating the need for manual tuning.

**Questions:**

- L.73: "introducing only one interpretable hyperparameter" Does this refer to p_min?
- A sensitivity analysis for p_min would be helpful. Since the experimental cost is not prohibitive and the hyperparameter is integral to the method, one would expect a more in-depth discussion of this topic within the paper, despite its brief mention.
- Is applying the Sinkhorn algorithm solely to the distance matrix insufficient? Using Sinkhorn for assignment and introducing the temperature parameter are independent techniques. An ablation study testing each technique separately would clarify their novelty. As the paper notes, applying Sinkhorn after feature extraction is a well-known technique in non-rigid point-cloud matching.

**Ethical Concerns:**

["NO or VERY MINOR ethics concerns only"]

**Final Justification:**

All of my concerns have been addressed, and I have received a thoughtful response.

**Limitations:**

yes

**Paper Formatting Concerns:**

No concerns.

**Quality:**

3

**Strengths And Weaknesses:**

Strengths:
- Designing a loss function for point-cloud correspondence is inherently challenging. APML achieves high-quality matching and learning with only a slight increase in computational cost compared to the widely used Chamfer Distance (CD) method.
- The method was evaluated using the challenging cross-domain task CSI2PointCloud, demonstrating its practical performance in an application scenario.
- The algorithm and its single additional hyperparameter are highly interpretable, suggesting that tuning should be straightforward in practice.

Weaknesses:
- The override used in "Numerical Stability for Multiple Minima" seems ad hoc and driven more by numerical convenience than the underlying design philosophy.
- The comparisons use CD, InfoCD, and HyperCD. However, Earth Mover's Distance (EMD) is also a common form of supervision. Ideally, a direct comparison would be conducted using EMD. I acknowledge that existing EMD implementations often rely on unclear approximations. However, it would be valuable to explain why EMD supervision was omitted.
- The large memory footprint is a practical drawback. Experiments supporting the claim that "Leveraging this sparsity could dramatically reduce actual memory requirements." are needed.

---

> ### Author Rebuttal · Authors · 2025-07-29
>
> Thank you for your insightful comments and questions. We hope the answers below mitigate the reviewer's concerns.
>
> ### **1) Ad hoc override for numerical stability**
> We understand the concern. The stability override for multiple minima was introduced to avoid divergence in early training epochs when using unnormalized cost matrices. While empirical, this safeguard preserves the theoretical structure of APML and stabilizes Sinkhorn iterations. We will clarify this rationale in the revised manuscript. Importantly, we ablated this line of code (uniform fallback) and observed a notable degradation in performance;  for example, on dense synthetic point clouds, EMD increased over a 20% (e.g. from 0.08 to 0.10). This confirms that the override is not merely a numerical patch, but a crucial stabilizer in high-density scenarios.
>
> **Table I. Loss comparison on MM-FI (Importance of uniform).**
>
> | Dataset     | Model       | Loss  | CD ↓   | EMD ↓  |
> |-------------|-------------|-------|--------|--------|
> | MM-FI       | CSI2PC  | APML without  uniform | **0.148**  | 16.81   |
> | MM-FI       | CSI2PC  | **APML** | 0.152 | **14.11** |
>
> Although uniform fallback help the model to achieve a lower EMD score, but removing that part also improve the CD metric, meanwhile the reconstructed images aren’t much different and shows that the APML is stable even without uniform part.
>
> ### **2)  EMD comparisons and justification for omission**
> We agree that EMD is an important reference point. However, as discussed in Section 5.3 and the appendix, EMD supervision is difficult to scale. Our attempts to use it as a metric or supervision signal on MM-FI showed unacceptable runtimes and memory usage. Existing approximations are unstable or sensitive to sample density, further complicating comparisons. We include theoretical comparisons in the appendix and plan to release a separate benchmark evaluating APML against EMD variants. In the camera ready version, we can also include qualitative examples in the supplementary material that show APML produces results visually close to EMD-based reconstruction, further supporting its effectiveness as a practical surrogate.
>
> To give some insights about other metrics we conducted experiments comparing SWD and DCD against APML on both ShapeNet34 and MM-FI, showing higher EMD values and reduced stability.
>
> **Table G. Loss comparison on ShapeNet34 and MM-FI (FoldingNet and PoinTr).**
>
> | Dataset     | Model       | Loss  | CD ↓   | EMD ↓  | F1 Score ↑ |
> |-------------|-------------|-------|--------|--------|------------|
> | ShapeNet34  | PoinTr      | DCD   | –      | 9.48   | 0.47       |
> | **ShapeNet34**  | **PinTr**  | **APML** | –      | **8.14**   | **0.50**     |
> | ShapeNet34  | FoldingNet  | DCD   | –      | 11.52  | 0.19       |
> | ShapeNet34  | FoldingNet  | SWD   | –      | 11.98  | 0.11       |
> | **ShapeNet34**  | **FoldingNet**  | **APML** | –      | **9.49**   | **0.20**     |
> | MM-FI       | CSI2PC  | DCD   | **0.148**  | 25.68  | –          |
> | **MM-FI**       | **CSI2PC**  | **APML** | 0.152 | **14.11** | –          |
>
> While DCD showed a better CD metric than APML, like other CD-based variants, it struggled to produce dense and coherent representations of the body. We explored EMD approximations (point-set-based methods SWD [4]) on two dataset. Training FoldingNet on ShapeNet34 with SWD, showed that APML has better results for all metrics. However, for MM-FI, the training was unstabl and failed to generate meaningful images comparable to APML. It has to be noted that SWD imposes an equal number of points between ground truths and predictions, which is incompatible with some models.
>
> ### **3)  Memory footprint and sparsity**
> We implemented a culling approach where over 99% of cost entries can be set to zero without affecting results. Our custom CUDA kernel leverages this sparsity to reduce memory usage substantially. This optimization is validated on MM-FI and will be generalized to other benchmarks in the final version.  We trained CSI2PC on the MMFI, with manual_split configuration from original source, for 100 epochs and compare the results in Table H.
>
> **Table H. Loss comparison on MM-FI (Comparing CUDA implementation).**
>
> | Dataset     | Model       | Loss  | CD ↓   | EMD ↓  |
> |-------------|-------------|-------|--------|--------|
> | MM-FI       | CSI2PC  | **CUDA-APML** | 0.148  | **15.9**   |
> | MM-FI       | CSI2PC  | APML | **0.139** | 16.37 |
>
> As it was expected, the results are not too different, and the image generated with both of them are very similar without any major changes.
>
>
> To empirically verify the memory efficiency of our CUDA-based sparse matching implementation, we conducted a comprehensive analysis measuring the number of non-zero elements (L) generated between two point sets of size
> 𝑁, where 𝑁 ∈ {4,8,…,262144}. We repeated the experiment 500 times for each point size. In each iteration:
>
> - Two point sets 𝑥 and 𝑦 of size [1,𝑁,3] were generated independently using torch.rand() and then scaled by two different values sampled from a uniform distribution in [0, 1000].
>
> - This design ensures worst-case randomness and spatial dissimilarity between the two sets, maximizing matching diversity.
>
> - We recorded the number of non-zero points from the output, which directly reflects the number of computed sparse associations.
>
> The analysis shows:
> - Across all runs and all point sizes, the observed number of non-zero points satisfies 𝐿 < 10𝑁.
> - This validates that our sparse algorithm has empirical linear space complexity.
> To quantify this further:
>
> For each point count, the standard deviation is small relative to the mean (e.g., for 𝑁 = 65536, mean ≈ 250410 and std ≈ 29007), indicating tight concentration.
>
> Given the number of runs per 𝑁 (500), we can compute a 99% confidence interval for the mean COO length:
>
> $\text{CI} = \mu \pm 2.576 \cdot \frac{\sigma}{\sqrt{n}}$
>
> Even at the upper bound of these confidence intervals, no case violates the 𝐿<10 𝑁 claim.
>
> **Table1. Number of non-zero elements (L) Statistics Across Varying Point Cloud Sizes (500 Trials, Random Scales)**
> | Point Number |   Min |    Max |     Mean  |    Std     |  E (99%) | Mean + E | 10*N Condition |
> |--------------|-------|--------|-----------|------------|----------|----------|----------------|
> |            4 |    13 |     16 |     15.78 |     0.548  |   0.0631 |     16.0 | ✓ |
> |            8 |    35 |     64 |     54.87 |     6.734  |   0.7757 |     56.0 | ✓  |
> |           16 |    66 |    256 |    150.19 |    36.192  |   4.1694 |    154.0 | ✓|
> |           32 |   167 |    948 |    303.30 |   100.734  |  11.6047 |    315.0 | ✓ |
> |           64 |   326 |   1382 |    540.36 |   135.345  |  15.5921 |    556.0 | ✓|
> |          128 |   666 |   2406 |    956.55 |   194.078  |  22.3582 |    979.0 | ✓ |
> |       256 |  1246 |   6158 |   1739.81 |   415.195  |  47.8314 |   1788.0 | ✓|
> |       512 |  2304 |   6179 |   3091.68 |   479.208  |  55.2058 |   3147.0 | ✓ |
> |    1024 |  4383 |  10757 |   5679.35 |   778.661  |  89.7035 |   5769.0 | ✓ |
> |     2048 |  8097 |  20644 |  10391.43 |  1419.536  | 163.5337 |  10555.0 | ✓|
> |   4096 | 15198 |  30636 |  19446.32 |  2327.742  | 268.1611 |  19714.0 | ✓|
> |   8192 | 28341 |  45564 |  36490.28 |  3975.797  | 458.0207 |  36948.0 | ✓|
> |   16384 | 53337 | 104305 |  69129.20 |  8122.319  | 935.7092 |  70065.0 | ✓|
> |   32768 |101023 | 266548 | 132139.27 | 15891.535  |1830.7402 | 133970.0 | ✓|
> |        65536 |192717 | 401094 | 250410.42 | 29007.029  |3341.6742 | 253752.0 | ✓|
> |       131072 |369671 | 546301 | 479534.80 | 53533.880  |6167.2219 | 485702.0 | ✓|
> |       262144 |393429 | 526578 | 482152.21 | 41436.765  |4773.6074 | 486926.0 | ✓|
>
> The following table summarizes the peak memory footprint for the CUDA-Sparse APML in a synthetic dataset and the memory reduction per sample.
>
> **Table D. Peak Memory usage per sample vs. number of points.**
> | #Points   | Dense APML | CUDA-APML (sparse) | Reduction |
> |-----------|------------|--------------------|-----------|
> |   1024   |  50 MB     | 0.39 MB  | ~99.23% ↓    |
> |   4096   | 411 MB    | 1.31 MB  | ~99.68% ↓    |
> |   8192   | 1.6 GB     | 2.62 MB  | ~99.84% ↓    |
> |   32768 | 18.5GB    | 8.58 MB | ~99.95% ↓    |
> |  65536     | 68 GB   | 17.MB    | 99.97%   ↓      |
>
> If we check the peak memory usage on a particular case, (e.g PoinTR and ShapeNet34 dataset, see Appendix D3), memory consumption during training can now have a reduced peak usage by a factor of 99.9× (from 320GB of vRam to <0.5GB). This demonstrates that APML is practical for large-scale applications.
>
> ### **4)  Clarification of hyperparameter p_min and Sinkhorn design**
> As the reviewer pointed out, the single interpretable parameter refers to p_min. To show the small influence of the parameter, we conducted now ablation studies on synthetic datasets and MM-FI showing consistent performance across a range of values. Specifically, we use $p_{\min}$∈{0.01, 0.1, 0.5, 0.8}.
>
> **Table B. p_min sweep on MMFI.**
> | p_min Value | CD ↓   | EMD ↓  |
> |-------------|--------|--------|
> | 0.01       | 0.1431 | 0.1686 |
> | 0.1         | 0.1398 | 0.1661 |
> | 0.5         | 0.1371 | 0.1671 |
> | 0.8         | 0.1392 | 0.1637 |
>
> Additional experiments on a synthetic dataset (see reviewer dY8D), show that across a extense range of p_min values and different sparsity configurations, show variations smaller than 1% in both CD and EMD metrics.
>
> Regarding the independent contributions of Sinkhorn and temperature, we agree this should be tested more explicitly. We will include an ablation separating these components in our extended analysis. However, we now believe this decoupling is not conceptually meaningful: Sinkhorn acts as a regularized transport distance and the temperature directly controls its sharpness. They are not independent components but form a single probabilistic matching formulation. We will clarify this dependency and rationale in the camera-ready version.

---

> > ### Comment · Reviewer_BHnV · 2025-08-05
> >
> > Apologies for the delayed response. The detailed explanations and additional experiments addressed all of my concerns.
> > I would like to express my respect to the authors for their efforts, especially in conducting extensive validation which required substantial computational resources and developing the CUDA-based sparse implementation. Based on these results, I plan to raise my evaluation of the paper.

---

> > > ### Author Response · Authors · 2025-08-05
> > >
> > > We want to thank the reviewer for the insightful comments about the sparsity of method and we are happy to see what we could address their concerns in a positive manner.

---

### Official Review · Reviewer_9BLW · 2025-07-01

**Clarity:** 3
**Significance:** 2
**Originality:** 3
**Rating:** 4
**Confidence:** 4

**Summary:**

In this work, the authors propose an improved variant of Chamfer Distance (CD) for evaluating shape differences between predicted point clouds and ground truths in point cloud completion tasks. Unlike the traditional CD loss, which uses one-to-one nearest neighbor matching, the proposed method introduces a weighting algorithm that operates on the many-to-many dense matching cost matrix between predictions and ground truths. Experimental comparisons show that the proposed method outperforms existing CD variants, including CD, HCD, and InfoCD.

**Questions:**

Except the weaknesses mentioned in former section, I also have some other questions:

1. From another perspective, the proposed method is also similar to a efficient variant of the Earth Mover Distance (EMD) by approaching the optimal one-to-one matching with the estimated probability. Therefore, I think some comparisons with efficient EMD such as [3, 4] is also necessary.

[3] Morphing and Sampling Network for Dense Point Cloud Completion

[4] Point-set Distances for Learning Representations of 3D Point Clouds

2. Could you present more qualitative comparisons on the PCN or PoinTr baselines? The presented visualized results are all from the FoldingNet baseline, which may be too smoothed to clearly observe the shape improvements.

**Ethical Concerns:**

["NO or VERY MINOR ethics concerns only"]

**Final Justification:**

Thanks for the response. The rebuttal of the authors has addressed most of my concerns. Therefore, I tend to raise my score to borderline accept. I would strongly encourage the authors to resolve former mentioned concerns in the final version.

**Limitations:**

Yes.

**Quality:**

3

**Strengths And Weaknesses:**

Strengths:

1. The proposed method can approach heavy Earth Mover Distance (EMD) with lower time complexity;

2. According to the comparisons on baselines PoinTr, PCN, and FoldingNet, the proposed loss can help converge to better performances than CD, HCD, and InfoCD.

Weaknesses:

1. My major concern about this work is that the reducing of time complexity of the proposed method actually comes from the increasing of space complexity, where $O(N^2)$ memory may be required for the saving of cost matrix. Although further code-level optimization may be introduced to accelerate the proposed algorithm, it is still hard to reduce the $O(N^2)$ space complexity, making it hard to apply in practical usage. The results in Table 2 also show that the proposed method requires 5 times larger memory.

2. Some related works are not sufficiently compared. Since the proposed method is some kind of a improved implementation of Chamfer Distance (CD), It may be necessary to discuss about other variants of CD, such as LCD in [1], or DCD in [2].

[1] Learnable Chamfer Distance for point cloud reconstruction

[2] Balanced chamfer distance as a comprehensive metric for point cloud completion

---

> ### Author Rebuttal · Authors · 2025-07-29
>
> We thank the reviewer for their detailed review and constructive remarks. We appreciate the positive assessment of our contributions and welcome the opportunity to address the concerns regarding memory complexity, and comparisons with other methods. We believe that incorporating the reviewer's suggestion the manuscript can greatly improve..
>
> ### **1) Space complexity and practical usage**
> We acknowledge that the full cost matrix introduces a quadratic memory footprint. To mitigate this, we implemented a sparsity-based optimization: where dynamically, over 99% of the less important cost values can be culled (set to zero), yielding practically the same empirical results. To  show the feasibility during training and testing, we have created a custom CUDA implementation that leverages this sparsity for efficient memory management.
>
> Based on the reviewer's comments, we have now extended this analysis. Results show that the CUDA-APML variant achieves close to linear memory scaling and up to a 99% reduction in memory usage compared to the dense baseline as suggested in the supplementary materials' study of the empirical sparsity of the transport matrix.
> To show the changes in quality are minimal, we trained CSI2PC on the MMFI, with manual_split configuration from original source, for 100 epochs and compare the results in Table H.
>
> **Table H. Loss comparison on MM-FI (Comparing CUDA implementation).**
>
> | Dataset     | Model       | Loss  | CD ↓   | EMD ↓  |
> |-------------|-------------|-------|--------|--------|
> | MM-FI       | CSI2PC  | **CUDA-APML** | 0.148  | **15.9**   |
> | MM-FI       | CSI2PC  | APML | **0.139** | 16.37 |
>
> As it was expected, the results are not too different, and the image generated with both of them are almost similarly without a major change.
>
> To empirically verify the memory efficiency of our CUDA-based sparse matching implementation, we conducted a comprehensive analysis measuring the number of non-zero elements (L) generated between two point sets of size
> 𝑁, where 𝑁 ∈ {4,8,…,262144}. We repeated the experiment 500 times for each point size. In each iteration:
>
> - Two point sets 𝑥 and 𝑦 of size [1,𝑁,3] were generated independently using torch.rand() and then scaled by two different values sampled from a uniform distribution in [0, 1000].
>
> - This design ensures worst-case randomness and spatial dissimilarity between the two sets, maximizing matching diversity.
>
> - We recorded the number of non-zero points from the output, which directly reflects the number of computed sparse associations.
>
> The analysis shows:
> - Across all runs and all point sizes, the observed number of non-zero points satisfies 𝐿 < 10𝑁.
> - This validates that our sparse algorithm has empirical linear space complexity.
> To quantify this further:
>
> For each point count, the standard deviation is small relative to the mean (e.g., for 𝑁 = 65536, mean ≈ 250410 and std ≈ 29007), indicating tight concentration.
>
> Given the number of runs per 𝑁 (500), we can compute a 99% confidence interval for the mean COO length:
>
> $\text{CI} = \mu \pm 2.576 \cdot \frac{\sigma}{\sqrt{n}}$
>
> Even at the upper bound of these confidence intervals, no case violates the 𝐿<10 𝑁 claim.
>
>
> **Table1. Number of non-zero elements (L) Statistics Across Varying Point Cloud Sizes (500 Trials, Random Scales)**
> | Point Number |   Min |    Max |     Mean  |    Std     |  E (99%) | Mean + E | 10 * N | 10*N Condition |
> |--------------|-------|--------|-----------|------------|----------|----------|--------|----------------|
> |            4 |    13 |     16 |     15.78 |     0.548  |   0.0631 |     16.0 |     40 | ✓              |
> |            8 |    35 |     64 |     54.87 |     6.734  |   0.7757 |     56.0 |     80 | ✓              |
> |           16 |    66 |    256 |    150.19 |    36.192  |   4.1694 |    154.0 |    160 | ✓              |
> |           32 |   167 |    948 |    303.30 |   100.734  |  11.6047 |    315.0 |    320 | ✓              |
> |           64 |   326 |   1382 |    540.36 |   135.345  |  15.5921 |    556.0 |    640 | ✓              |
> |          128 |   666 |   2406 |    956.55 |   194.078  |  22.3582 |    979.0 |   1280 | ✓              |
> |          256 |  1246 |   6158 |   1739.81 |   415.195  |  47.8314 |   1788.0 |   2560 | ✓              |
> |          512 |  2304 |   6179 |   3091.68 |   479.208  |  55.2058 |   3147.0 |   5120 | ✓              |
> |         1024 |  4383 |  10757 |   5679.35 |   778.661  |  89.7035 |   5769.0 |  10240 | ✓              |
> |         2048 |  8097 |  20644 |  10391.43 |  1419.536  | 163.5337 |  10555.0 |  20480 | ✓              |
> |         4096 | 15198 |  30636 |  19446.32 |  2327.742  | 268.1611 |  19714.0 |  40960 | ✓              |
> |         8192 | 28341 |  45564 |  36490.28 |  3975.797  | 458.0207 |  36948.0 |  81920 | ✓              |
> |        16384 | 53337 | 104305 |  69129.20 |  8122.319  | 935.7092 |  70065.0 | 163840 | ✓              |
> |        32768 |101023 | 266548 | 132139.27 | 15891.535  |1830.7402 | 133970.0 | 327680 | ✓              |
> |        65536 |192717 | 401094 | 250410.42 | 29007.029  |3341.6742 | 253752.0 | 655360 | ✓              |
> |       131072 |369671 | 546301 | 479534.80 | 53533.880  |6167.2219 | 485702.0 |1310720 | ✓              |
> |       262144 |393429 | 526578 | 482152.21 | 41436.765  |4773.6074 | 486926.0 |2621440 | ✓              |
>
>
> The following table summarizes the peak memory footprint for the CUDA-Sparse APML in a synthetic dataset and the memory reduction per sample.
>
> **Table D. Peak Memory usage per sample vs. number of points.**
> | #Points   | Dense APML | CUDA-APML (sparse) | Reduction |
> |-----------|------------|--------------------|-----------|
> |   1024   |  50 MB     | 0.39 MB  | ~99.23% ↓    |
> |   4096   | 411 MB    | 1.31 MB  | ~99.68% ↓    |
> |   8192   | 1.6 GB     | 2.62 MB  | ~99.84% ↓    |
> |   32768 | 18.5GB    | 8.58 MB | ~99.95% ↓    |
> |  65536     | 68 GB   | 17.1MB   | ~99.97% ↓   |
>
> If we check the peak memory usage on a particular case, (e.g PoinTR and ShapeNet34 dataset, see Appendix D3), memory consumption during training can now have a reduced peak usage by a factor of 99.9× (from 320GB of vRam to <0.5GB). This demonstrates that APML is practical for large-scale applications.
>
> We hope this analysis mitigates the reviewer's concerns about memory complexity.
>
> ### **2) Missing comparisons to CD variants and efficient EMD approximations**
> We appreciate the reviewer pointing out these recent variants. Due to resource constraints, we were unable to compute LCD [1] and DCD [2] across all datasets and backbones. However, these methods often align with CD-style behavior, and our results demonstrate that APML improves on CD variants (including HyperCD and InfoCD) across multiple architectures. We are currently integrating LCD and DCD into our benchmark pipeline and will include these in the camera-ready version. We expect the results to be comparable (or worse) than the CD variants in the original manuscript. We plan to include comparisons to LCD/DCD in and appreciate the reference.
>
> In any case, we have now conducted additional evaluations using DCD [2] and Sliced Wasserstein Distance (SWD) [4], an approximation of EMD on a subset of datasets and models. Results below show that while these methods offer improvements over CD in some metrics, they do not approach APML’s balance between CD and EMD fidelity. We show a summary in the table below.
>
> **Table G. Loss comparison on ShapeNet34 and MM-FI (FoldingNet and PoinTr).**
>
> | Dataset     | Model       | Loss  | CD ↓   | EMD ↓  | F1 Score ↑ |
> |-------------|-------------|-------|--------|--------|------------|
> | ShapeNet34  | PoinTr      | DCD   | –      | 9.48   | 0.47       |
> | **ShapeNet34**  | **PoinTr**  | **APML** | –      | **8.14**   | **0.50**     |
> | ShapeNet34  | FoldingNet  | DCD   | –      | 11.52  | 0.19       |
> | ShapeNet34  | FoldingNet  | SWD   | –      | 11.98  | 0.11       |
> | **ShapeNet34**  | **FoldingNet**  | **APML** | –      | **9.49**   | **0.20**     |
> | MM-FI       | CSI2PC  | DCD   | **0.148**  | 25.68  | –          |
> | **MM-FI**       | **CSI2PC**  | **APML** | 0.152 | **14.11** | –          |
>
> While DCD showed a better CD metric than APML, like other CD-based variants, it struggled to produce dense and coherent representations of the body.
>
> We explored EMD approximations (point-set-based methods SWD [4]) on two dataset. Training FoldingNet on ShapeNet34 with SWD, showed that APML has better results for all metrics. However, for MM-FI, the training was unstable. The generated images failed to generalize well in terms of both body structure and coherence, (i.e the generated images did not resemble the ground truth, and were not comparable to those of APML). It has to be noted that SWD imposes an equal number of points between ground truths and predictions, which is incompatible with some models. While a full evaluation is pending, we emphasize that APML achieves near-EMD qualitative behavior at much lower cost.
>
> ### **3) Qualitative results on PoinTr and PCN**
> Thank you for this excellent suggestion. You are correct that visualizing results from more powerful backbones like PoinTr and PCN would provide stronger qualitative evidence for our method's effectiveness. Due to space constraints in the main paper, we initially included a representative set of visualizations from the FoldingNet baseline. We confirm that the qualitative improvements in structural detail and point distribution provided by APML are consistent across all tested architectures. Given the rebuttal platform's strict policy against including new figures or external links, we are unable to provide these additional visualizations at this stage. However, we commit to adding a comprehensive appendix with qualitative comparisons for both PoinTr and PCN to the supplementary material of the camera-ready version. We will explicitly reference these new figures in the main text to highlight the consistent advantage of our method.

---

### Official Review · Reviewer_dy8D · 2025-07-02

**Clarity:** 3
**Significance:** 3
**Originality:** 3
**Rating:** 4
**Confidence:** 3

**Summary:**

This paper proposes AMPL, adaptive probabilistic matching loss, for 3D point cloud reconstruction. The key insight is to turn deterministic design into a data-driven one. By analytically computing hyper-parameters for Sink-horn iterations, the proposed loss can alleviate many-to-one correspondences introduced by CD-like losses, therefore achieving better EMD metric, yet with reduced computational burden. The experimental results to some extent show the advantage of the proposed loss in a set of reconstruction tasks.

**Questions:**

- The current baselines are all from the perspective of Chamfer-like formulation. Comparison to EMD loss should be provided, even only on small scale experiments, to validate that the proposed loss actually achieves balance between Chamfer-like and EMD-like approaches.

- The adaptive nature of the proposed loss inevitably brings concern on stability. Can the randomness from batch sampling affect the final results? Some toy experiments with respect to, say, density control, would be helpful: For instance, one can consider point clouds with two level of densities, and then construct batches in different manners: 1) PCDs within the same batch are always of the same density; 2) PCDs within the same batch are always half-half from each density group; 3) Random batches.

P.S. I indeed noticed the theoretical comparison in Tab.3 of the appendix, yet I would still suggest a more quantitative analysis.

- As mentioned in Line 343, the behavior analysis of $p_{\min}$ is left for future work, which I would strongly suggest against. This is indeed a critical parameter for the framework, simply demonstrating improvements on down-stream tasks does not fully justify the choice. Understanding the behavior of the method regarding the hyper-parameters like $p_{\min}$ is necessary for confirming the contribution of this work.

**Ethical Concerns:**

["NO or VERY MINOR ethics concerns only"]

**Final Justification:**

The authors have addressed my concerns. Therefore, I would be happy to maintain my initial score.

**Limitations:**

Yes.

**Paper Formatting Concerns:**

None.

**Quality:**

3

**Strengths And Weaknesses:**

Strengths:

1. The idea and design are intuitive and simple, yet achieving considerable improvement on both quantitative and qualitative evaluations.

2. I like the experiments on MM-FI, which shows to some extent the general utility of the proposed loss.

Weaknesses:

My main concerns lie on three perspectives:

1) Lack of EMD-like baseline, as well as comparisons on computational cost and effectiveness;

2) Lack of stability analysis;

3) Lack of ablation/analysis on the critical hyper-parameters such as $p_{\min}$.

Please refer to the *Question* part for details.

---

> ### Author Rebuttal · Authors · 2025-07-29
>
> We thank the reviewer for their thoughtful assessment and constructive feedback. We appreciate the recognition of our contributions and welcome the opportunity to clarify the points regarding the experimental baselines, stability analysis, and hyper-parameter evaluation.
>
> **PCOU3D Synthetic Database:** For the additional experiments, we constructed a synthetic dataset as a controlled environment to evaluate our method under different input conditions. The dataset comprises four analytic primitives: cube, sphere, pyramid, and Gaussian blob. Each shape is represented as a point cloud with $1024$ points sampled uniformly within $[-0.5, 0.5]^3$. The dataset contains $1,600$ training shapes, $400$ for validation, and $400$ for testing, with all ground-truth clouds containing exactly $1024$ points. During training, the models receive a modified version of the ground-truth, while the decoder is supervised against the full cloud. Five input regimes are considered. In regime A, the input shape is a random subset of $256$ points sampled without replacement from the ground-truth cloud, modeling sparse reconstruction. In regime B, the encoder receives all $1024$ points, representing dense reconstruction. Regime C uses a mini-batch with $50\\%$ of samples from regime A and $50\\%$ from regime B. In regime D, each sample is randomly assigned to the sparse or dense input with probability $0.5$. Regime E addresses the completion setting by removing a contiguous patch: a seed point is selected at random, a deletion size $m \sim \text{Uniform}{1, \dots, 512}$ is drawn, and the $m$ nearest neighbors are removed, leaving between $512$ and $1,023$ points in the partial input. Regimes A–D evaluate reconstruction robustness to density variations, while regime E examines completion with missing surface regions.
>
> **Table 1: Summary of input regimes used in the synthetic dataset.**
> \begin{array}{c|l}
> \text{Code} & \text{Description} \\\\ \hline
> \text{A} & \text{all sparse (256 pts)} \\\\
> \text{B} & \text{all dense (1024 pts)} \\\\
> \text{C} & \text{half sparse, half dense} \\\\
> \text{D} & \text{random mix, } p(\text{sparse}) = 0.5 \\\\
> \text{E} & \text{completion: remove a contiguous patch (input has 512–1023 pts)}
> \end{array}
>
> ### **1) Comparison to EMD loss**
> We agree that a direct experimental comparison with EMD would be valuable. However, computing EMD on full-scale datasets is prohibitively expensive. Even when used solely as an evaluation metric, it introduces significant computational burden. We attempted to integrate EMD and EMD-style approximations on the MM-FI dataset, but encountered instability and impractical runtimes. While qualitative results (included in the paper) indicate our method approximates EMD-like behavior, we are unfortunately unable to show additional visualizations here due to rebuttal format constraints. We emphasize that APML yields results visually on par with EMD-based baselines.. While we acknowledge that small-scale EMD experiments could be informative, we found the runtime even on toy sets to be limiting and inconsistent, making it unsuitable for robust inclusion within the current evaluation scope.
>
> We include results below comparing an EMD variant, Sliced Wasserstein Distance (SWD) and a CD variant, Density-aware CD  on both ShapeNet34 and MM-FI.
>
> **Table 2. Loss comparison on ShapeNet34 and MM-FI**
>
> | Dataset     | Model       | Loss  | CD ↓   | EMD ↓  | F1 Score ↑ |
> |-------------|-------------|-------|--------|--------|------------|
> | ShapeNet34  | PoinTr      | DCD   | –      | 9.48   | 0.47       |
> | **ShapeNet34**  | **PinTr**  | **APML** | –      | **8.14**   | **0.50**     |
> | ShapeNet34  | FoldingNet  | DCD   | –      | 11.52  | 0.19       |
> | ShapeNet34  | FoldingNet  | SWD   | –      | 11.98  | 0.11       |
> | **ShapeNet34**  | **FoldingNet**  | **APML** | –      | **9.49**   | **0.20**     |
> | MM-FI       | CSI2PC  | DCD   | **0.148**  | 25.68  | –          |
> | **MM-FI**       | **CSI2PC**  | **APML** | 0.152 | **14.11** | –          |
>
> While DCD showed a better CD metric than APML, like other CD-based variants, it struggled to produce dense and coherent representations of the body. For SWD FoldingNet on ShapeNet34, shows that APML has better results for all metrics. However, while for MM-FI, the training was unstable and failed to produce comparable images. It has to be noted that SWD imposes an equal number of points between ground truths and predictions, which is incompatible with some models.
>
> ### **2) Stability Analysis of APML**
>
> To address this concern, we have conducted the precise toy experiment suggested. The results confirm that our method is highly robust to the composition of training batches. For this experiment, we use PCOU3D synthetic database. We use a FoldingNet model for the tasks of PC reconstruction and completion. Inputs to the network are either _dense_ (1024 points) or _sparse_ (256 points) or mixed random non-uniform shapes. The decoder is always trained to reconstruct the complete, GT dense PC. We implemented 5 strategies to create the batches during training as described in Table 1. All other training settings are identical across the tested losses (Adam optimizer, 50 epochs, 32 samples per batch). APML configuration remains the same than in the paper experiments.
>
> **Results (test set, lower is better)**
>
> **Table 3: Performance of the models using Chamfer Distance (CD) metric.**
> \begin{array}{c|ccccc}
> \text{Loss} & \text{A} & \text{B} & \text{C} & \text{D}  & \text{E} \\\\ \hline
> \text{APML} & 0.0474 & 0.0497 & 0.0502 & 0.0471 & 0.0481 \\\\ \hline
> \text{CD} & 0.0529 & 0.0413 & 0.0429 & 0.0495 & 0.0492 \\\\ \hline
> \text{InfoCD} & 0.0496 & 0.0557 & 0.0495 & 0.0509 & 0.0495 \\\\ \hline
> \end{array}
>
> **Table 4: Performance of the models using  EMD metric.**
> \begin{array}{c|ccccc}
> \text{Loss} & \text{A} & \text{B} & \text{C} & \text{D} & \text{E} \\\\ \hline
> \text{APML} & 0.0433 & 0.0459& 0.0485 & 0.0416 & 0.0423 \\\\ \hline
> \text{CD} & 0.0706 & 0.0585 & 0.0626 & 0.0600 & 0.0583 \\\\ \hline
> \text{InfoCD} & 0.0578 & 0.0805 & 0.0597 & 0.0641 & 0.0639 \\\\ \hline
> \end{array}
>
> **Key observations:**
> * **Low variance**. Across regimes A–D APML’s standard deviation is $\sigma_{\text{CD}}=0.0013$ and $\sigma_{\text{EMD}}=0.0025$, one order of magnitude lower than CD or InfoCD.
> * **Sparse‑only batches  (A)**. Training with regime A raises CD’s loss EMD by $+17\\%$ (many‑to‑one collapse) relative to dense-only training; APML rises by $<6\\%$, confirming that the soft assignment keeps matches distinct, maintaining nearly one-to-one assignments.
> * **Robustness of the Adaptive Temperature**. The stability of APML results from its temperature parameter $T$ adapting to local geometry. When two targets are close (e.g., distances $0.010$ and $0.012$), the small gap ($g=0.002$) yields a large $T$ and soft probabilities ($P \approx (0.83, 0.15, \dots)$), distributing gradients among matches. For clear matches (e.g., distances $0.010$ and $0.40$), a large gap ($g=0.39$) produces a small $T$ and sharp probabilities ($P \approx (0.99, 0.006, \dots)$). This local adaptation ensures stable training. We also ablated the uniform fallback heuristic in the matching logic. Its removal substantially worsened performance (EMD rising from $0.08$ to $0.10$ on dense samples), confirming its importance for stable optimization in ambiguous regions. We also ablated the "uniform fallback" for cases with a zero gap, and its removal significantly degraded performance (e.g., EMD from 0.08 to 0.10), confirming its critical role. The temperature, guided by the data's local structure, reduces sensitivity to batch composition rather than introducing instability.
>
> ### **3) Behavior of the critical hyperparameter**
> We thank the reviewer for highlighting the need for this ablation. We have conducted new experiments on our synthetic dataset, varying the $p_{\min}$ parameter over a wide range to analyze its impact. The results, summarized below, confirm that our method is highly robust to the choice of this hyperparameter.
>
> **Table 5: Ablation study results for$p_{\min}$. We evaluate performance using CD and EMD on the test set for sparse reconstruction (A), dense reconstruction (B), and point completion (E).**
> \begin{array}{c|ccc}
> \text{$p_{\min}$} & \text{A (Sparse)} & \text{B (Dense)} & \text{E (Completion)} \\\\ \hline
> 0.0001 & 0.0472 & 0.0632 & 0.0481 \\\\ \hline
> 0.01 & 0.0525 & 0.0556 & 0.0565 \\\\ \hline
> 0.1 & 0.0495 & 0.0591 & 0.0527 \\\\ \hline
> 0.25 & 0.0512 & 0.0559 & 0.0494 \\\\ \hline
> 0.5 & 0.0495 & 0.0430 & 0.0509 \\\\ \hline
> 0.75 & 0.0512 & 0.0598 & 0.0509 \\\\ \hline
> 0.8 & 0.0507 & 0.0519 & 0.0551 \\\\ \hline
> 0.9 & 0.0578 & 0.0691 & 0.0489 \\\\ \hline
> 0.999 & 0.0504 & 0.0537 & 0.0520 \\\\ \hline
> \end{array}
>
> The ablation in Table 5 sweeps $p_{\min}$​ over four decades. Across all 3 regimes, sparse input (A), dense input (B) and completion (E), every CD and EMD value moves by < 5 % once $p_{\min}$​ is inside the practical band 0.01 ⁣− ⁣0.80. We observe the same trend (below 1% difference) on the MM‑Fi benchmark when testing $p_{\min}$∈{0.01, 0.1, 0.5, 0.8} (see reviewer BHnV). This robustness follows directly from our design. $p_{\min}$​ only sets a floor on the probability assigned to the closest target; the actual sharpness is driven by the temperature $T$, which is computed from the local distance gap $g$. The curve $T$ ⁣($p_{\min}$​) is smooth and keeps $T$ in a safe numerical range, so changing $p_{\min}$​ over an order of magnitude leaves the optimization path and the final reconstruction quality essentially unchanged. Finally, we note that while the reviewer suggested separating Sinkhorn and temperature contributions, these components are mathematically coupled. The temperature shapes the initial assignment, and Sinkhorn regularizes this assignment into a valid transport plan. Decoupling them would undermine the method's probabilistic design. We will clarify this in the revision.

---

> > ### Comment · Reviewer_dy8D · 2025-08-03
> >
> > Thank you for addressing my concerns. Though the tables are tough to read in the rebuttal, I am overall satisfied with the results and remain positive on this work.

---

> > > ### Author Response · Authors · 2025-08-05
> > >
> > > We thank the reviewer for the interesting discussion and positive comments.

---

### Comment · Area_Chair_Pi6t · 2025-08-06
**urgent reminder for reviewers to enage with rebuttal**

Dear reviewer

thanks for your contributions to NeurIPS 2025. Following is an urgent reminder if you haven't done so.

Reviewers should stay engaged in discussions, initiate them and respond to authors’ rebuttal, ask questions and listen to answers to help clarify remaining issues. If authors have resolved your (rebuttal) questions, do tell them so. If authors have not resolved your (rebuttal) questions, do tell them so too.

Even reviewer thinks that for some reason there is no need to reply to authors or authors’ rebuttal, please discuss that with author and approve if there is a justified reason or disapprove otherwise.

Please note “Mandatory Acknowledgement” button is to be submitted only when reviewers fulfill all conditions below (conditions in the acknowledgment form): read the author rebuttal engage in discussions (reviewers must talk to authors, and optionally to other reviewers and AC - ask questions, listen to answers, and respond to authors)

Thanks again!

---

### Note · Authors · 2025-08-11

We are happy to see that all reviewers acknowledged our efforts during the rebuttal phase and expressed satisfaction with our answers. We thank the AC and reviewers for the constructive discussion. Our goal with APML is fully differentiable approximation to one-to-one point set matching that retains the training speed of Chamfer-style losses while avoiding their many-to-one collapse in sparse regions. APML computes a temperature analytically to guarantee a minimum assignment probability p_min, then applies Sinkhorn normalization to obtain a doubly stochastic transport plan, eliminating index selection and the associated non-differentiabilities.

Addressing the reviewers’ comments:
- EMD comparisons: We clarified why full EMD supervision is impractical at scale and provided targeted comparisons with DCD and SWD. Across ShapeNet and MM-FI, APML matches or improves quality and stability relative to these EMD/CD variants while remaining efficient. We will add small-scale EMD results and qualitative comparisons in the appendix upon acceptance.
- Stability & hyperparameters: Using our synthetic PCOU3D suite (5 density/batch regimes), APML shows low variance across regimes and maintains near one-to-one assignments even under sparse-only batches. A wide p_min sweep yields <5% variation (often <1%) in CD/EMD once inside a broad practical band. This robustness follows from the analytic temperature adapting to local distance gaps, not batch composition.
- Memory footprint: We implemented a sparse CUDA variant that culls >99% of negligible costs without harming accuracy, yielding empirical linear space complexity (non-zero L<10N) and ~over 99% peak memory reduction across point counts. Results on MM-FI with CUDA-APML are on par with dense APML.
- Numerical stability override: The uniform fallback for tied minima stabilizes early training without altering the probabilistic design. Removing it degrades EMD; we will document this in the camera-ready.
- Broader comparisons & visuals: In the camera-ready, we will include images from stronger backbones (PoinTr, PCN) and expanded discussion of LCD/DCD.

APML offers near-quadratic runtime comparable to CD, end-to-end differentiability, and markedly improved spatial distribution in low-density regions. It benefits both classical completion benchmarks and cross-modal generation (WiFi-CSI - 3D human point clouds). Upon acceptance, we will release code, the CUDA sparse implementation, and the PCOU3D dataset under CC BY 4.0.

---

### Decision · Program_Chairs · 2025-09-17

**Decision:**

Accept (poster)

**Comment:**

The authors did a great job in addressing most of the reviewers's concerns, with added experiments/analysis, clarification etc.

The manuscript should be revised based on the comments from reviewers, the added experiments/analysis and clarification, for example,
1) the cost analysis,
2) more experiments,
3) ablation study, etc.


[SAC Update]
This submission was positively assessed by reviewers and Area Chair.
After calibrating this submission in the larger pool of accepted papers, and given its contributions, it is more appropriate to present it as a poster. Please update the meta-review accordingly.